# Polyglutamylation of microtubules drives neuronal remodeling

Antoneta Gavoci [1], Anxhela Zhiti [1], Michaela Rusková[2,3],
Maria M. Magiera [4,5], Mengzhe Wang[1], Karin A. Ziegler [6,7],
Torben J. Hausrat [8], Anselm I. Ugwuja [1], Shreyangi Chakraborty[4,5],
Stefan Engelhardt [6,7], Matthias Kneussel [8], Martin Balastik [2],
Carsten Janke [4,5], Thomas Misgeld [1,9,10,11] & Monika S. Brill [1,10,11] ✉

Developmental remodeling shapes neural circuits via activity-dependent pruning of synapses and axons. Regulation of the cytoskeleton is critical for this process, as microtubule loss via enzymatic severing is an early step of pruning across many circuits and species. However, how microtubule-severing enzymes, such as spastin, are activated in specific neuronal compartments remains unknown. Here, we reveal that polyglutamylation, a post-translational tubulin modification enriched in neurons, plays an instructive role in developmental remodeling by tagging microtubules for severing. Motor neuron-specific gene deletion of enzymes that add or remove tubulin polyglutamylation—TTLL glutamylases vs. CCP deglutamylases—accelerates or delays neuromuscular synapse remodeling in a neurotransmission-dependent manner. This mechanism is not specific to peripheral synapses but also operates in central circuits, e.g., the hippocampus. Thus, tubulin polyglutamylation acts as a cytoskeletal rheostat of remodeling that shapes neuronal morphology and connectivity.

Microtubules are regulators of cellular shape, dynamics, and transport processes that are especially difficult to coordinate in the complex cells of the nervous system[1–3]. Consequently, microtubule functions in neurons need to be locally regulated. In analogy to how post-translational modifications (PTMs) diversify histone function, the tubulin code concept posits that microtubules can be endowed with unique functions through PTMs and a set of proteins that add, remove, or interpret them. Such a system of writer, eraser, and reader microtubule-associated proteins (MAPs) and enzymes can then locally modify microtubule stability, as well as the trafficking characteristics and cargo selection of intracellular transport with high spatiotemporal precision[3–6]. Thus, in principle, the tubulin code can explain how a general system of filamentous proteins could locally regulate a plethora of parallel functions to meet the compartmentalized needs of neural cells. Indeed, the microtubule cytoskeleton of neurons is very specialized. This specialization includes the expression of neuron-specific tubulin isotypes (such as tubulin beta-3, Tubb3), as well as a unique set of MAPs and severing enzymes that play essential roles in neurodegenerative and neurodevelopmental diseases[7–11]. A further PTM-based refinement of the tubulin code is an appealing concept. Implementations of such a system have been shown to play crucial roles in neurons' basic health and cell biology, e.g., via modifying cell

[1]Institute of Neuronal Cell Biology, Technical University of Munich, Munich, Germany. [2]Department of Molecular Neurobiology, Institute of Physiology of the Czech Academy of Sciences, Prague, Czech Republic. [3]Faculty of Sciences, Charles University, Prague, Czech Republic. [4]Institut Curie, Université PSL, Orsay, France. [5]Université Paris-Saclay, Orsay, France. [6]Institute of Pharmacology and Toxicology, Technical University of Munich, Munich, Germany. [7]German Centre for Cardiovascular Research (DZHK), Partner Site Munich Heart Alliance, Munich, Germany. [8]Department of Molecular Neurogenetics, Center for Molecular Neurobiology, ZMNH, University Medical Center Hamburg-Eppendorf, Hamburg, Germany. [9]German Center for Neurodegenerative Diseases (DZNE), Munich, Germany. [10]Munich Cluster of Systems Neurology (SyNergy), Munich, Germany. [11]These authors jointly supervised this work: Thomas Misgeld, Monika S. Brill. ✉e-mail: monika.leischner-brill@tum.de

migration, axonal transport, growth cone navigation, and cilia function[3,5,12–16]. Still, how the tubulin code could generally steer nervous system physiology in vivo, particularly circuit development, has only begun to be understood.

Synaptic pruning—the widespread removal of exuberant synapses in an activity-dependent manner, a process preserved across phyla and between development and disease—involves local regulation of cytoskeletal function and stability[17–20]. For instance, others and we have previously described the selective loss of microtubules in destabilized presynaptic axon branches[17,21,22]. This loss can be mediated by the microtubule-severing enzyme, spastin[21], a protein commonly altered in hereditary spastic paraplegia[9]. Indeed, microtubule destabilization, together with glia-mediated engulfment of synaptic material, is thus far one of the very few shared mechanisms of synapse pruning across models and between development and disease[23,24]. The local loss of microtubules in pruning axon branches is accompanied by changes in PTMs[21]. However, whether these cytoskeletal changes are instructive or merely an epiphenomenon of the breakdown, which molecular signals locally regulate microtubule modifications, and whether activity impinges on them, remains unknown. To address this question, we focus on polyglutamylation, a tubulin PTM[25] highly enriched in the brain[26]. Tubulin polyglutamylation involves the enzymatic seeding of an initial glutamate followed optionally by adding further glutamate residues, forming a polyglutamate (PolyE) chain. This PTM enhances the negative charge of the microtubule lattice, thus facilitating electrostatic interactions with neurodegeneration-associated MAPs, such as tau[27–29], and severases, including spastin[30,31].

Based on these prior observations, we systematically investigated the role of microtubule polyglutamylation during pruning. For this, we took advantage of the unique features of synaptic pruning at the neuromuscular junction (NMJ). In mice, this synapse transitions from innervation by multiple to a single axon branch within a few days. Due to its accessibility and size, the NMJ allows unique dynamic investigations with subsynaptic resolution[21,32–34], which we used to reveal an instructive role of tubulin polyglutamylation in synaptic remodeling. Specifically, we show that in motor axon branches during pruning, polyglutamylation of tubulin alpha-4A (Tuba4a) results from the balanced action of the writer enzyme, tubulin tyrosine ligase-like glutamylase 1 (TTLL1; but not the functionally distinct TTLL7[13]) and the eraser enzymes, cytosolic carboxy peptidases (CCP) 1 and 6. The local density of polyglutamyl side chains on microtubules, modified in cell type-specific knock-out mice and altered by blocking activity of developing NMJs, determines the efficiency of spastin-mediated severing. We found that this interplay of PTMs and microtubule severing paces the speed by which peripheral synapses remodel and confirmed a related mechanism for the timing of axon and spine pruning in the developing hippocampus.

## Results
### Glutamylases and deglutamylases are translationally regulated during motor axon pruning

To establish the expression patterns of polyglutamylation writer and eraser enzymes during motor axon pruning, we performed a translatome analysis of murine motor neurons using a ChAT-IRES-Cre X Rpl22[HA] (called RiboTag hereafter)[35,36] mouse cross-breeding. We immuno-isolated ribosome-bound mRNA from spinal cords, starting at postnatal day (P) 5 until 14 (P5, 7, 9, 11, 14; Fig. 1a). Specificity of this experiment for motor neuronal transcripts was verified by immunostaining for HA, which illustrated clear co-localization with Choline Acetyl Transferase (ChAT) (Fig. 1b), and quantitative PCR, which demonstrated a ≈20x enrichment of *ChAT* and absence of Glial Fibrillary Acidic Protein (*GFAP*) transcripts in the pull-downs compared to whole spinal cord mRNA (Supplementary Fig. 1a, b). Principal component (PCA) analysis separated the RiboTag pull-down vs. the total spinal cord mRNA samples (Supplementary Fig. 1c). In total, we

identified >4500 differentially expressed transcripts at P9 (RiboTag vs whole spinal cord; $Log_2 |FC| \geq 1.5$; $P_{adj} \leq 0.05$; Supplementary Data 1). Gene ontology (GO) analysis of this gene list pointed towards neuron- or axon-specific biological processes, including ones involving the microtubule cytoskeleton (Supplementary Fig. 1d, e). Gene set enrichment analysis further confirmed augmented motor neuron markers (*Isl1, Chat, Mnx1*). In contrast, astrocyte markers (*GFAP, Slc1a3, Aldh1l1, Olig2, Cspg4*) and oligodendrocyte-specific genes (*Sox10, Olig2, Cspg4*) were depleted in the RiboTag pull-down samples (Supplementary Fig. 1f). We then scrutinized the complete gene list (Supplementary Data 2) obtained from our bona fide motor neuron translatome for glutamylases (*TTLL1, 3, 4, 5, 6, 7, 11* and *13*) and deglutamylases (*CCP1, 2, 3, 4, 5* and *6*; Supplementary Data 2).

Writer TTLL glutamylases decorate the microtubule lattice with glutamate either in an initiator fashion by preferentially adding the first branched glutamate (E) onto the tubulin C-terminal tail (monoE) or by elongating previously initiated E-chains[13,37] (Fig. 1c). Among the known elongators, *TTLL1* showed the highest expression levels (e.g., at P14: approximately 69% of all elongator *TTLL* reads) that were stable across all analyzed time points. In contrast, *TTLL11* was expressed at a substantially lower level (e.g., at P14: 29%), while *TTLL6* and *TTLL13* were virtually absent (Fig. 1d). Analysis of initiators yielded increasing *TTLL7* mRNA reads between P5 and P14 (e.g., at P14: 87% of initator TTLL levels), while *TTLL5* and *TTLL4* transcripts were much less abundant (at P14: 11% and 1% for *TTLL5* and *TTLL4*, respectively; Fig. 1e). Amongst the deglutamylases, *CCP1* transcripts were the most abundant and increased during the pruning phases (e.g., increase from P5 to P14: 53 %; Fig. 1f). Among the other deglutamylases (*CCP2, 3, 4, 5, 6*), *CCP6* was the most abundant enzyme which exhibited an increase of 28% between P5 and P14. The remaining CCPs were present at very low transcript rates, ranging, e.g., from 0.03% to 1.1% at P14 (Fig. 1f). These findings are consistent with previous reports showing that CCP1 and CCP6 are the two most abundant deglutamylases in the central nervous system[14,38]. Thus, we focused on the genetic disruption of *TTLL1, TTLL7, CCP1*, and *CCP6* to probe their role as potential candidates for mediating polyE-instructed neuronal remodeling. We first generated cholinergic neuron-conditional mouse mutants of the most expressed glutamylases, i.e., TTLL1[mnKO] (ChAT-IRES-Cre X TTLL1[flox/flox]; see experimental procedures), which should primarily affect polyglutamylation on alpha-tubulin[13], and TTLL7[mnKO] (ChAT-IRES-Cre X TTLL7[flox/flox]; see experimental procedures), which is expected to deplete monoglutamylation on beta-tubulin[13].

### Genetic ablation of TTLL1, but not of TTLL7, delays neuromuscular synapse elimination

We quantified microtubule content and PTM patterns by immunohistochemistry and confocal microscopy in the triangularis sterni muscles of these mice at P9[21,34,39]. To determine microtubule content, we used an antibody directed against tubulin beta-3 (Tubb3), the most abundant beta-tubulin isotype in motor neurons after P9 (Supplementary Fig. 1g), and an anti-alpha-tubulin antibody, recognizing all alpha-tubulin isotypes. In parallel, an anti-PolyE antibody, which recognizes C-terminal chains of ≥3 glutamate residues, was used to assess axonal polyglutamylation levels. As expected, PolyE normalized to Tubb3 was drastically reduced (to 0.09-fold) in TTLL1[mnKO] compared to TTLL1[mnWT], while microtubule content increased (by 2.8-fold and 1.7-fold for Tubb3 and Tuba, respectively; normalized to neurofilament). In contrast, neurofilament heavy polypeptide content was not significantly altered (Fig. 2a–h and Supplementary Fig. 2a–c). In contrast, immunostaining using an anti-βmonoE antibody (normalized to Tubb3), which detects a single glutamate seed on E435 of tubulin beta-2 tails, was largely unaffected (Fig. 2i, j and Supplementary Fig. 2d). Finally, antibody staining for acetylated tubulin (acetyl-K40) was also decreased (0.68-fold, normalized to Tubb3, Fig. 2k l and Supplementary Fig. 2e), in line with previous reports[40]. Thus, microtubules in

motor axons in TTLL1[mnKO] mice are more abundant but may carry fewer PTMs. As revealed by time-lapse imaging of nerve-muscle-explants of TTLL1[mnKO] that also had a Thy1-EB3-YFP transgene[41], most parameters of plus-end polymerization dynamics were unaffected (including comet density, orientation, and velocity, Supplementary Fig. 3a, c, and d). Still, comet length distribution was significantly, albeit mildly, altered in TTLL1[mnKO] compared to TTLL1[mnWT] mice (Supplementary Fig. 3b). Specifically, EB3 comet length in TTLL1[mnKO] increased, aligning with recent in vitro findings that show augmented microtubule growth rate upon deglutamylation[42]. Concomitant to the increased microtubule mass and reduced PolyE levels, a higher percentage of neuromuscular synapses remained innervated by two or more axons in TTLL1[mnKO] vs. control muscles at all postnatal ages tested (Fig. 2i, j). At the same time, the overall neuromuscular development seemed normal (Supplementary Fig. 4a–c). These findings suggest that TTLL1-mediated polyglutamylation instructs the local dismantling of microtubules and thus paces the developmental axon pruning of motor neurons.

In contrast, the same analyses in TTLL7[mnKO] mice revealed no changes in overall microtubule mass and dynamism in terminal motor axons (Tubb3 staining normalized on NF; Fig. 3a, b, Supplementary Fig. 3e–h and Supplementary Fig. 5a). In line with the specificity of TTLL7 activity[13], the intensity of PolyE immunostaining was not altered (normalized to Tubb3; Fig. 3c, d and Supplementary Fig. 5b). Moreover, quantification of βmonoE immunofluorescence revealed a decrease compared to control axons as expected (0.55-fold, normalized to Tubb3; Fig. 3e, f). Consistently, the pruning speed of motor axons was similar in TTLL7[mnKO] vs. control muscles (Fig. 3g and Supplementary Fig. 5c). These data corroborate that only the density of TTLL1-added polyglutamylation, but not TTLL7-catalyzed βmonoE seeds, influences pruning, suggesting a high level of specificity in the regulation of pruning via microtubule PTMs, in line with the tubulin code concept.

## TTLL1[KO] leads to pruning defects in the CNS

We next explored whether reduced TTLL1-mediated polyglutamylation of microtubules would also result in defective pruning processes in the CNS. We chose granule neurons of the dentate gyrus, which express TTLL1 (Supplementary Fig. 6a–d) and provide a well-established and easily quantifiable murine model for

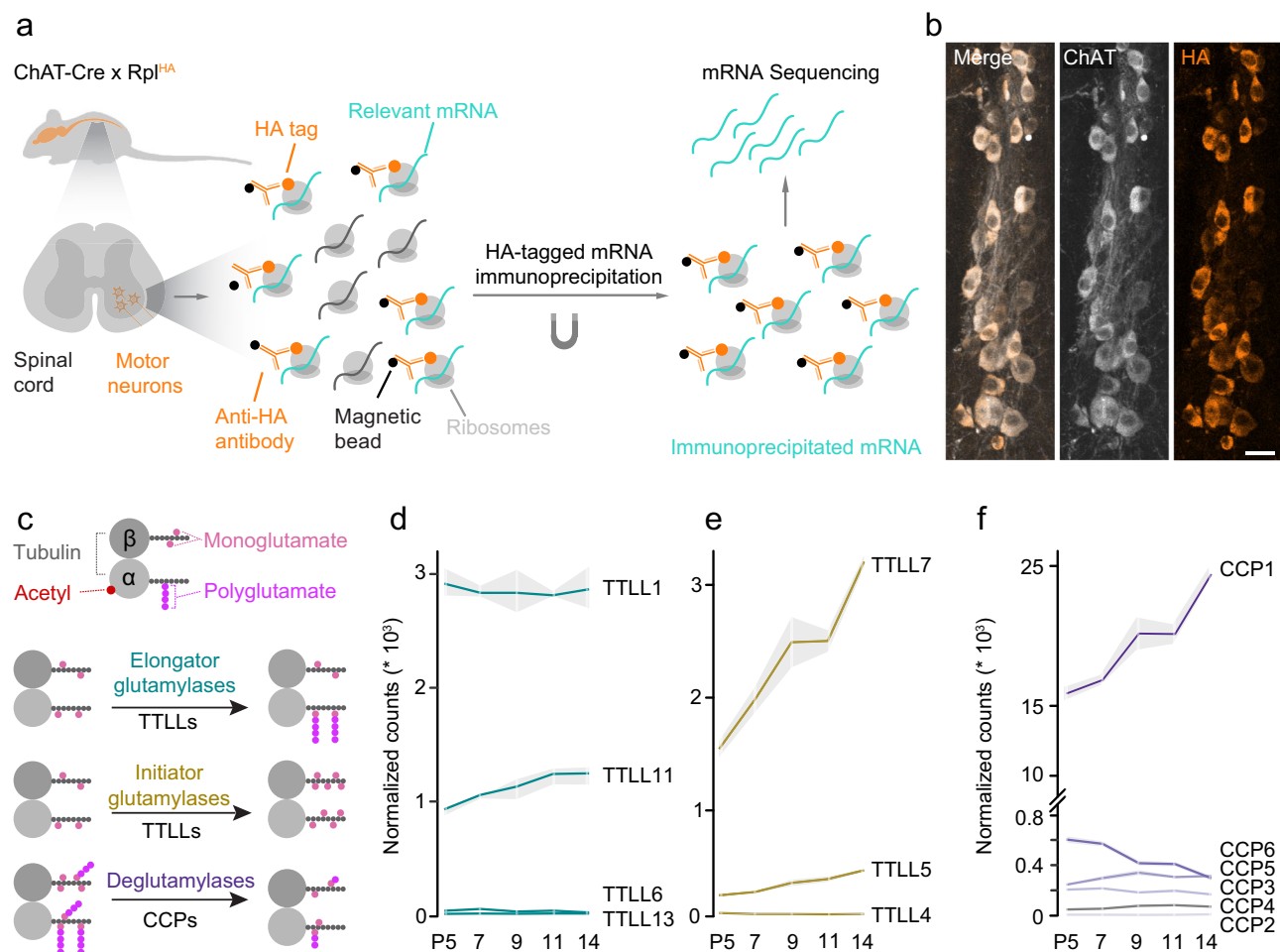

**Fig. 1 | Motor neuron transcriptome during postnatal remodeling. a** Strategy to isolate and sequence motor neuron-specific mRNA from ChAT-Cre mice crossbred to Rpl22[HA]. HA-tagged ribosomes (orange) and associated mRNA (cyan) were immunoprecipitated from spinal cords at different developmental time points (postnatal day, P5, 7, 9, 11, 14) and sequenced. **b** Confocal image stacks of longitudinal spinal cord sections of 8-week-old ChAT-Cre X Rpl22[HA] mice immunostained for choline acetyltransferase (ChAT, gray) and hemagglutinin (HA, orange), merged channels on the left (*n* = 3 mice). **c** Schematic illustration of tubulin post-translational modifications investigated in this study. Microtubules consist of alpha (α) and beta (β) tubulin dimers (gray), which may carry modifications in their luminal face (acetyl, red) or on their C-terminal tails (monoglutamate, pink; polyglutamate, magenta). Elongator TTLL glutamylases (turquoise) extend glutamyl side chains (polyglutamate, magenta); initiator TTLL glutamylases (gold) catalyze the addition of the first glutamate (monoglutamate, pink) to tubulin C-terminal tails; CCP deglutamylases (purple) remove glutamate residues. **d–f** Normalized mRNA read counts of (**d**) elongator *TTLL* glutamylases, (**e**) initiator *TTLL* glutamylases, and (**f**) *CCP* deglutamylases in motor neurons at postnatal day (P) 5, 7, 9, 11, 14 (*n* = 3 mice per age group). Graphs: Mean ± SEM. Scale bar, 25 μm.

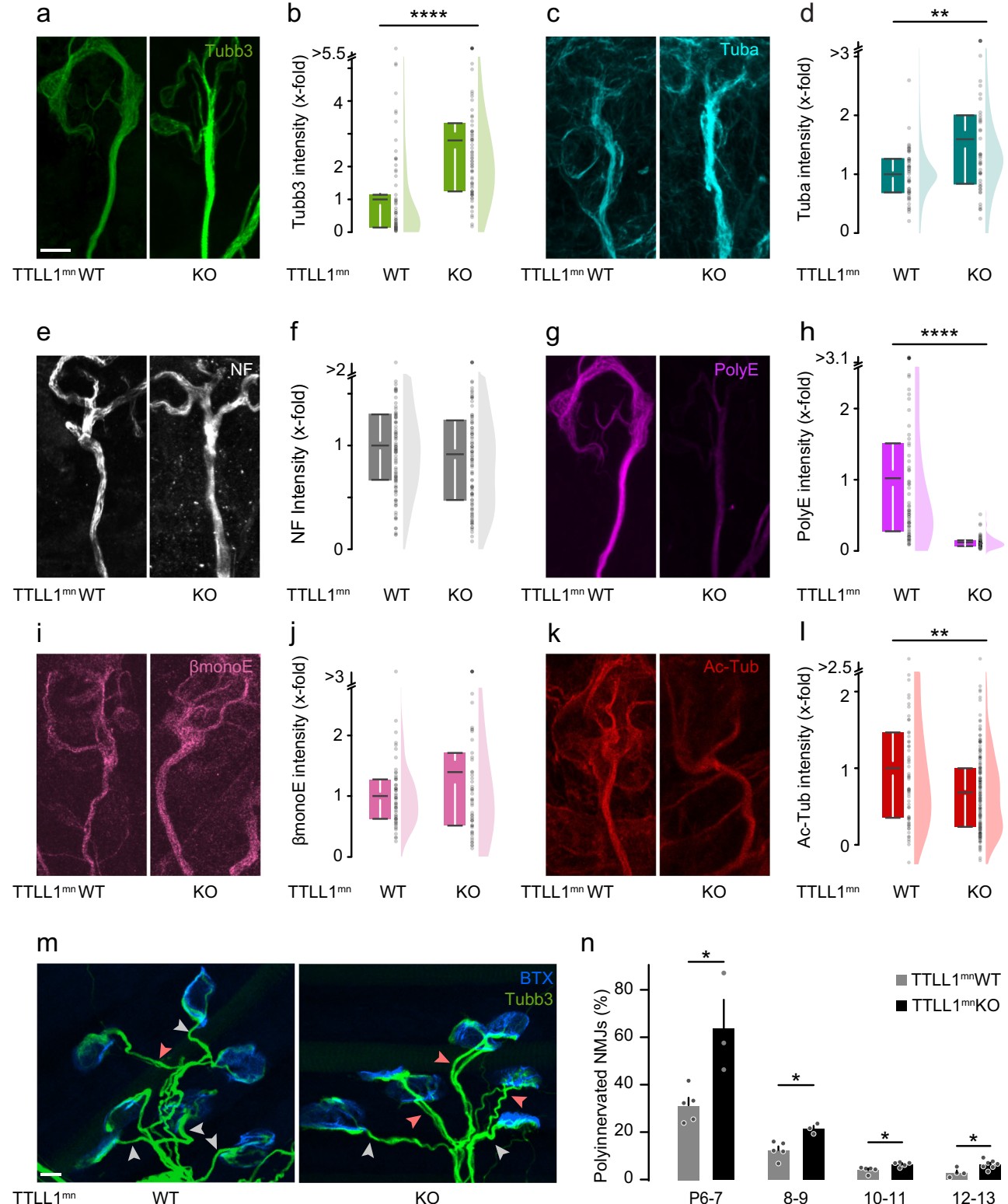

stereotyped pruning. During the first two postnatal months, granule cells dismantle their infrapyramidal bundle (IPB) of mossy fibers while the main suprapyramidal bundle (SPB) is maintained (see schematic; Fig. 4a). We used a constitutive TTLL1[KO] mouse model, which exhibits reduced PolyE levels in the brain[14]. Quantification of the IPB/SPB length ratio in sections immunostained for Calbindin D-28K, which selectively stains mossy fibers, revealed defective pruning of the IPB in TTLL1[KO] vs. control mice (Fig. 4b–d and Supplementary Fig. 7). We further tested if the elimination of spines in

granule neuron dendrites−i.e., remodeling of input synapses further upstream in hippocampal circuitry−was also impaired. Gene-gun based DiI labeling[43] before (3 week-old mice) and after the pruning phase (8 week-old mice) showed that TTLL1[KO] mice initially developed normal spine densities, but failed to prune (Fig. 4e−g). This suggests that TTLL1-mediated polyglutamylation might be a general pacemaker for the execution of remodeling processes, not only during competitive elimination of peripheral synapses but also during stereotyped pruning in the CNS.

**Fig. 2 | Genetic ablation of TTLL1 delays motor axon remodeling.**
**a**–**k** Quantitative immunostainings for microtubule markers and neurofilament heavy polypeptide on triangularis sterni muscles at postnatal day (P) 8–12 from TTLL1$^{mnKO}$ vs. TTLL1$^{mnWT}$ littermate controls. Confocal stacks of terminal axons leading to NMJs depicting immunostainings against (**a**) tubulin beta-3 (Tubb3, green), (**c**) alpha-tubulin (Tuba, cyan), (**e**) neurofilament (NF, white), (**g**) polyglutamate chains (PolyE, magenta), (**i**) monoglutamate on beta-tubulin (βmonoE, pink), and (**k**) acetylated tubulin (Ac-Tub, red). **b**–**l** Corresponding quantifications of immunostaining intensities in TTLL1$^{mnKO}$ terminal motor axons (right; relative to TTLL1$^{mnWT}$, left, which is set to 1). **b** Tubb3, normalized to NF (WT $n = 70$ axons, 3 mice; KO $n = 72$ axons, 3 mice, $p$-value = 7,04949E-13). **d** Tuba normalized to NF (WT $n = 46$ axons, 3 mice; KO $n = 50$ axons, 4 mice, $p$-value = 0.0036). **f** NF (WT $n = 108$ axons, 7 mice; KO $n = 100$ axons, 6 mice). **h** polyE normalized to Tubb3 (WT $n = 70$ axons, 3 mice; KO $n = 72$ axons, 3 mice, $p$-value = 1,23212E-24). **j** βmonoE, normalized to Tubb3 (WT $n = 58$ axons, 3 mice; KO $n = 54$ axons, 3 mice). **l** ac-tub,

normalized to Tubb3 NF (WT $n = 68$ axons, 4 mice; KO $n = 177$ axons, 8 mice, p-value = 0.00546). **m** Confocal stacks showing NMJs of P12 TTLL1$^{mnKO}$ and TTLL1$^{mnWT}$ littermates crossbred to Thy1-YFP or stained for Tubb3 (green; BTX, blue), depict singly (gray arrowheads) and polyinnervated synapses (light red arrowheads). **n** Percentage of polyinnervated NMJs in TTLL1$^{mnWT}$ vs. TTLL1$^{mnKO}$ littermates at postnatal day (P) 6-7, 8-9, 10-11, 12-13. (P6-7 WT $n = 5$, KO $n = 3$, $p$-value = 0.0357; P8-9 WT $n = 5$, KO $n = 3$, $p$-value = 0.0357; P10-11 WT $n = 5$, KO $n = 5$ mice, $p$-value = 0.0159, WT $n = 4$, KO $n = 7$, $p$-value = 0.0242; ≥ 197 NMJs per animal). Graphs: 25– 75% quantiles shown as a box with top and bottom black lines, mean as middle black line, and SEM as white lines (left); data representing single axons as dots (middle); half violin (right) in (**b**, **d**, **f**, **h**, **j**, **l**) or mean + SEM, with data representing single animals as dots in (**n**). A two-sided Mann-Whitney test determined significance: *, $P < 0.05$; **, $P < 0.01$; ****, $P < 0.0001$. Scale bar in (**a**) 5 μm also applies to (**c**, **e**, **g**, **i**, **k**), in (**m**) 10 μm. Source data are provided as a Source Data file.

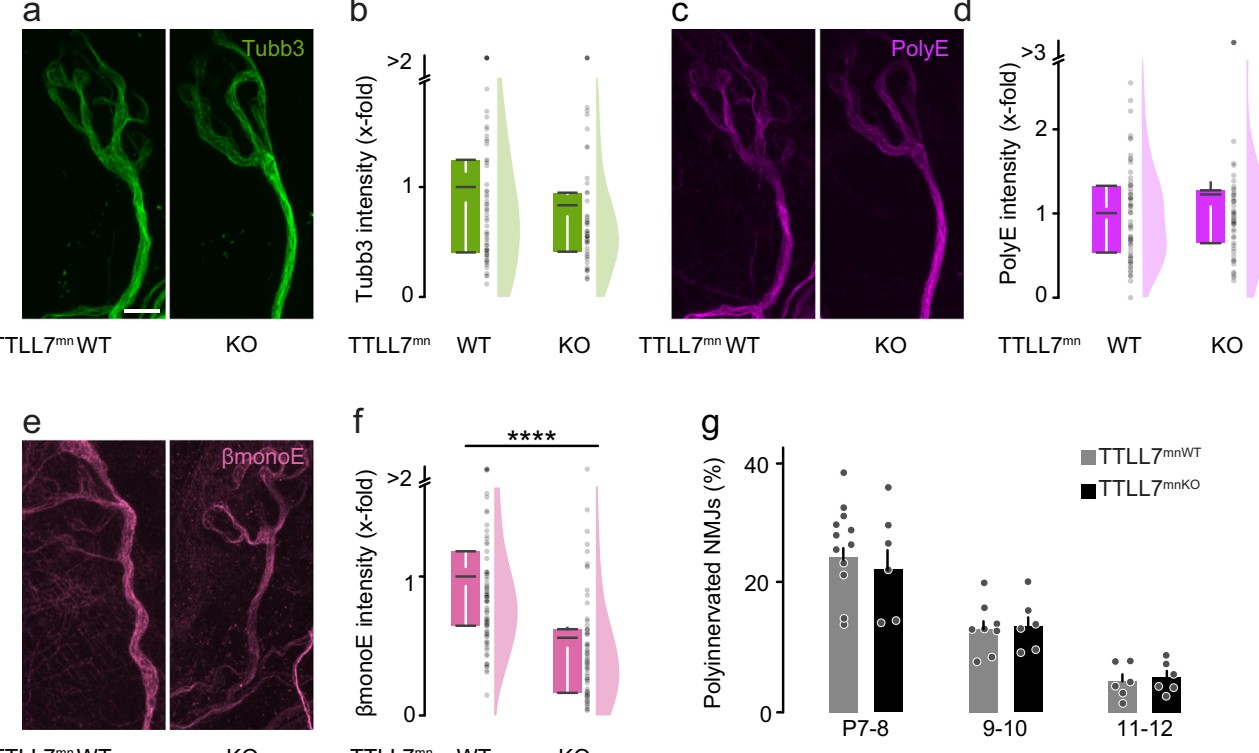

**Fig. 3 | Genetic ablation of TTLL7 does not affect peripheral pruning.**
**a**–**f** Quantitative immunostainings for microtubule markers and neurofilament heavy polypeptide on triangularis sterni muscles at P9 from TTLL7$^{mnKO}$ vs. TTLL7$^{mnWT}$ littermate controls. Confocal stack of a terminal axon leading to an NMJ showing (**a**) tubulin beta-3 staining (Tubb3, green), (**c**) polyglutamate chain staining (PolyE, magenta; same NMJ as in (**a**) and (**e**) beta-tubulin staining (βmonoE, pink). Corresponding quantifications of immunostaining intensities in TTLL7$^{mnKO}$ terminal motor axons (right; relative to TTLL7$^{mnWT}$, left, which is set to 1) of: (**b**) Tubb3, normalized to neurofilament heavy polypeptide (WT $n = 65$ axons, 5 mice, KO $n = 55$ axons, 4 mice), (**d**) polyE intensity, normalized to tubulin beta-3 (WT $n = 65$ axons, 5 mice, KO $n = 55$ axons, 4 mice) and (**f**) βmonoE, normalized to tubulin beta-3 (WT

$n = 94$ axons, 4 mice, KO $n = 76$ axons, 4 mice, $p$-value = 2,73259E-11). **g** Percentage of polyinnervated NMJs in TTLL7$^{mnWT}$ vs. TTLL7$^{mnKO}$ littermates crossbred to Thy1-YFP or stained for Tubb3 (green; BTX, blue) at postnatal day (P)7-8, 9-10, and 11-12 (P7-8 WT $n = 12$, KO $n = 6$; P9-10 WT $n = 8$, KO $n = 6$; P11-12 WT $n = 6$, KO 6 mice; ≥96 NMJs per animal). Graphs: 25–75% quantiles shown as a box with top and bottom black lines, mean as middle black line, and SEM as white lines (left); data representing single axons as dots (middle); half violin (right) in (**b**, **d**, **f**) or mean + SEM, with data representing animals as dots in (**g**). A two-sided Mann-Whitney test determined significance: ****, $P < 0.0001$. Scale bar, 5 μm in (**a**), applies also to (**c**) and (**e**). Source data are provided as a Source Data file.

## Polyglutamylation of tubulin alpha-4A instructs remodeling

Similar to previous reports about the brain[44], exploration of tubulin RNA levels in our motor neuron translatome data revealed an upregulation of tubulin alpha-4A (Tuba4a) among the alpha isotypes, with a consistent increase across the synaptic pruning phase (+63% from P5 to P14; Fig. 5a). In contrast, other alpha-tubulin isotypes were either stably expressed (e.g., Tuba1b and Tuba1c; Fig. 5a) or downregulated (Tuba1a; Fig. 5a). Interestingly, Tuba4a is the isotype that seems to carry the most extended glutamate side chains[45,46]. In light of this

expression pattern, we investigated the effect of Tuba4a-specific polyglutamylation loss in neuronal remodeling. We utilized Tuba4a knock-in mice (Tuba4a$^{KI}$), which carry a mutated Tuba4a allele prohibiting glutamylation of this specific tubulin isotype[45]. Using confocal analysis of double immunostainings for PolyE and Tubb3 in terminal axons of Tuba4a$^{KI}$ triangularis sterni muscles, we found a phenotype resembling that of TTLL1$^{mnKO}$. Tubb3 intensity was increased in Tuba4a$^{KI}$ vs. Tuba4a$^{WT}$ (1.2-fold; normalized to neurofilament, Fig. 5b, c and Supplementary Fig. 8a, b), while the intensity of the PolyE signal

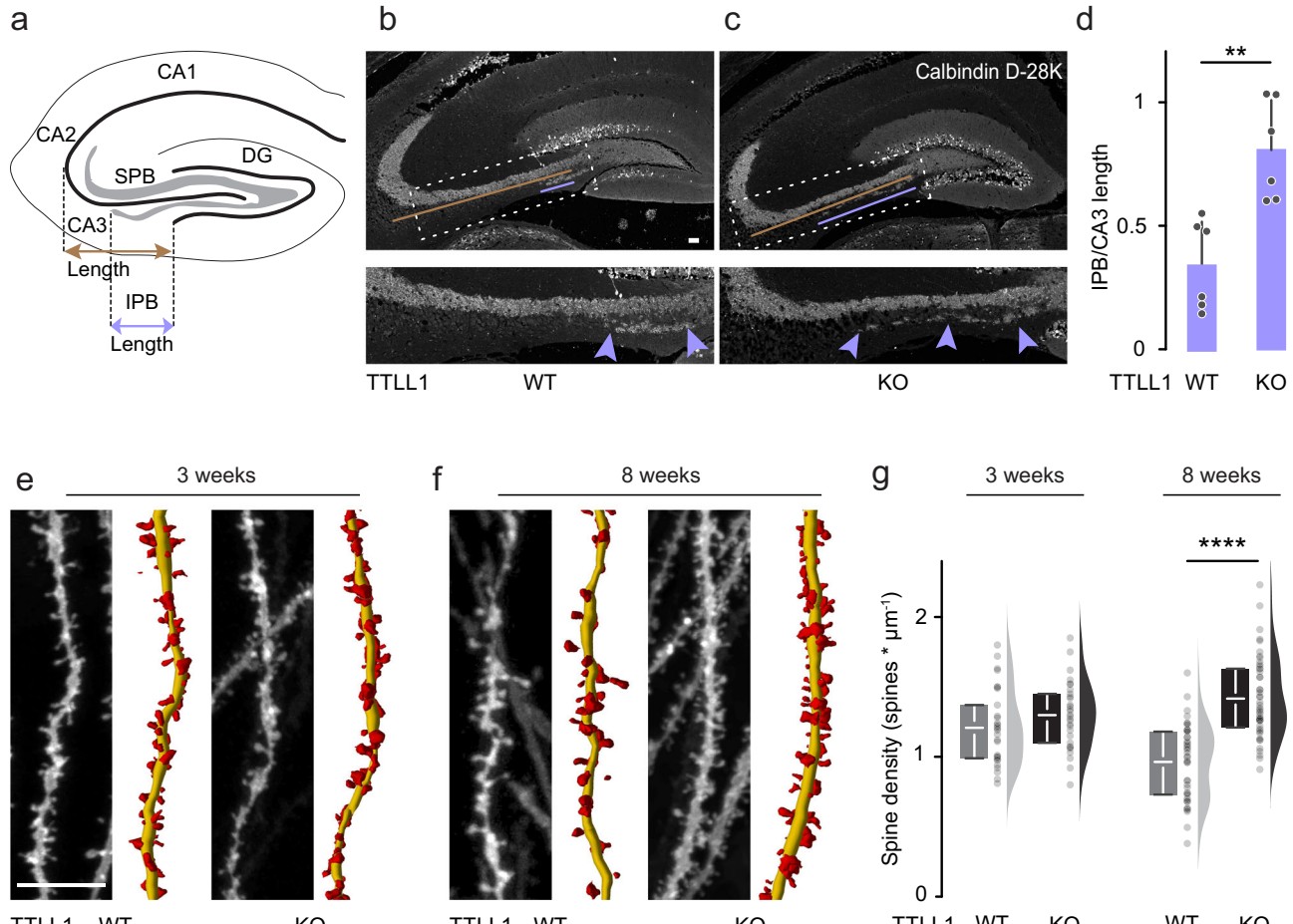

**Fig. 4 | TTL1$^{KO}$ leads to pruning defects in the CNS. a** Schematic of the hippocampus showing the infrapyramidal bundle (IPB, gray), the suprapyramidal main bundle (SPB, gray), the dentate gyrus (DG), as well as the CA1, CA2, CA3 regions; the violet line indicates IPB length and the brown line CA3 length, as used in quantification in (**d**). **b, c** Confocal stacks of coronal brain sections of 8-week-old TTLL1$^{WT}$ vs. TTLL1$^{KO}$ littermates immunostained for calbindin D-28k (white). The expansion of the dashed box shows IPB length at higher magnification (violet arrowheads); the violet line indicates IPB length, and the brown line CA3 length. **d** Quantification of IPB length normalized to CA3 length in 8-week-old TTLL1$^{WT}$ vs. TTLL1$^{KO}$ littermates ($n = 6$ hemispheres, 3 mice per genotype, $p$-value = 0.0022). **e, f** Confocal stacks of brain slices of TTLL1$^{WT}$ and TTLL1$^{KO}$ mice depicting DiI-labeled dendritic spines of hippocampal granule cells at (**e**) 3 (P21) and (**f**) 8 weeks (P56) of age. The right side panels illustrate spine reconstructions (dendrites, yellow; spines, red). **g** Quantification of dendritic spine density in TTLL1$^{WT}$ vs. TTLL1$^{KO}$ at 3 weeks (WT $n = 30$ dendrites, 3 mice, KO $n = 30$ dendrites, 3 mice, ≥8 dendrites per mouse, $p$-value = 0.1149) and 8 weeks (WT $n = 55$ dendrites, 4 mice, KO $n = 48$ dendrites, 4 mice, ≥9 dendrites per mouse, $p$-value = <0.0001). Graphs: mean + SEM and data represent hemispheres as single dots in (**d**); 25–75% quantiles as a box with top and bottom black lines, mean as middle black line, SEM as white lines (left); data representing dendrites as single dots (middle); half violin (right) (**g**). A two-sided Mann-Whitney test determined significance. **, $P < 0.01$; ****, $P < 0.0001$. Scale bars, 100 μm (**b, c**) and 10 μm (**e, f**). Source data are provided as a Source Data file.

staining was reduced (0.9-fold; normalized to Tubb3; Fig. 5d, e and Supplementary Fig. 8c). In line with polyglutamylation of tubulin alpha-4A being necessary to drive the pruning of motor axons initially, we found a significant albeit transient delay (Fig. 5f). At the same time, there was again no overt phenotype on neuromuscular development (Supplementary Fig. 4d–f).

### Genetic deletion of deglutamylases CCP1 and CCP6 in motor neurons accelerates pruning

We next wondered whether the described system operates in both directions, meaning that hyper-polyglutamylation of microtubules would accelerate axon pruning. Deglutamylases (CCPs) catalyze the removal of glutamate chains from tubulin C-terminal tails[38,47,48] and thus counteract the enzymatic activity of elongating glutamylases. We focused our efforts on combined motor neuron-specific deletions of CCP1 and CCP6 (CCP1&6$^{mnKO}$), despite CCP1 being the dominant CCP transcript in our translatome data (Fig. 1f), as CCP6 is known to have a compensatory potential[14].

Immunofluorescence analysis in terminal triangularis sterni axons at P9 revealed a reduction of microtubules, based on Tubb3 and Tuba levels, but not NF, in CCP1&6$^{mnKO}$ mice compared to littermate controls (0.8-fold; normalized to neurofilament; Fig. 6a–f and Supplementary Fig. 9a, b). Unexpectedly, PolyE immunostaining intensity was significantly decreased (0.6-fold; normalized to Tubb3; Fig. 6g, h; Supplementary Fig. 9c), while βmonoE levels remained unaltered as predicted (normalized to Tubb3; Fig. 6i, j and Supplementary Fig. 9d). In contrast to the TTLL1$^{mnKO}$ phenotype, acetylated microtubules were increased significantly (Fig. 6k, l; Supplementary Fig. 9e). Furthermore, we observed changes in EB3-YFP comet dynamics, with a 27% drop in density and a 12% increase in mean velocity in CCP1&6$^{mnKO}$ compared to control mice (Supplementary Fig. 3i, k; Supplementary Movie 1). Microtubule plus-end growth (EB3 comet orientation) and length distribution remained normal (Supplementary Fig. 3j, l; Supplementary Movie 1). Correlative live-imaging of EB3-YFP comet density in terminal axons normalized to tubulin beta-3 immunostaining in fixed tissue on the single axon level was unchanged in the CCP1&6$^{mnKO}$ compared to

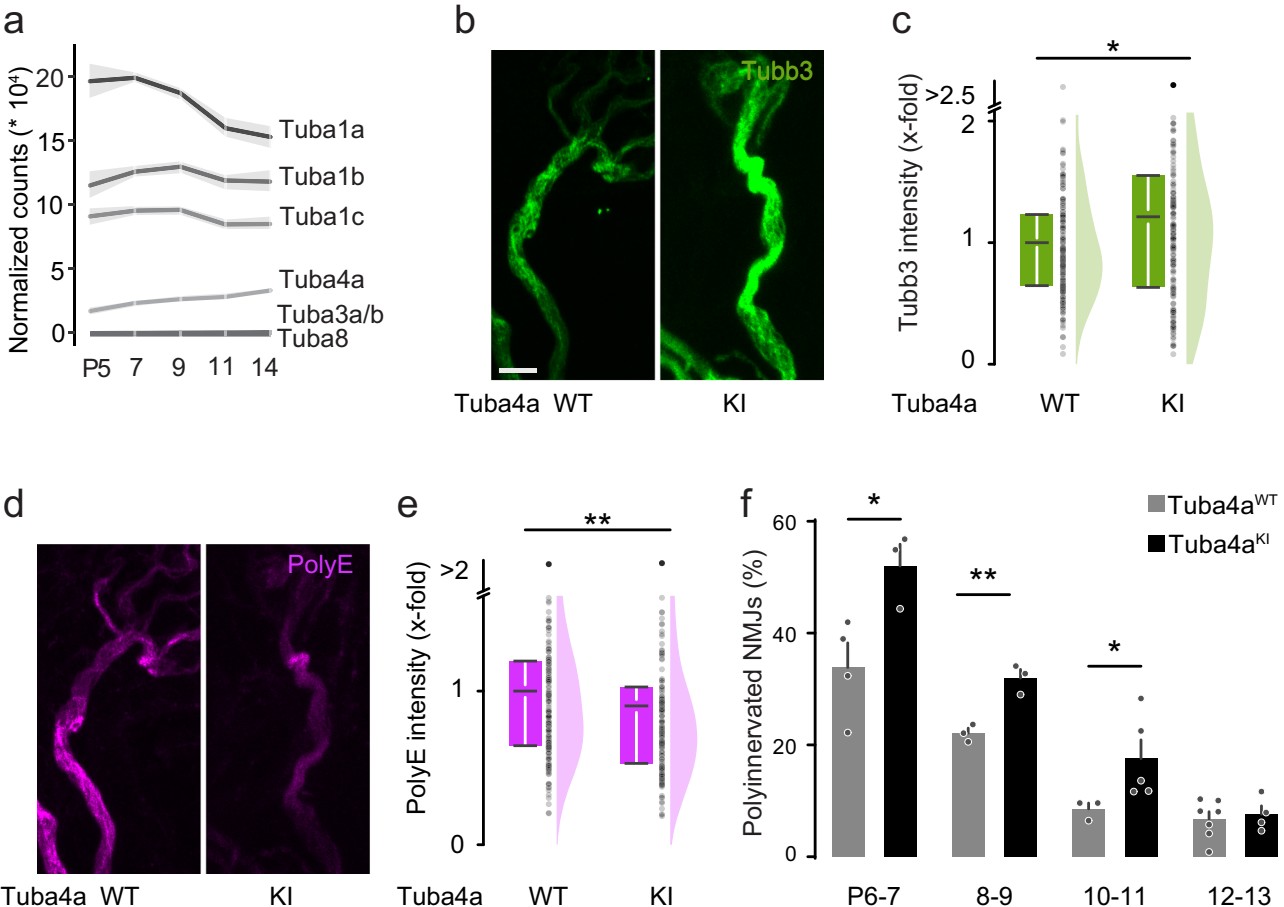

**Fig. 5 | Polyglutamylation of tubulinα4A instructs remodeling. a** Graph depicting normalized mRNA counts of alpha-tubulin isotypes across the postnatal motor axon remodeling phase (postnatal day (P) 5, 7, 9, 11, 14). **b**–**e** Quantitative immunostainings for tubulin beta-3, neurofilament heavy polypeptide, and polyglutamate on triangularis sterni muscles at P10-13 from Tuba4a[KI] vs. Tuba4a[WT] littermate controls. Confocal stacks of terminal axons leading to NMJs depicting immunostainings against (**b**) tubulin beta-3 (Tubb3, green) and (**d**) polyglutamate chains (PolyE, magenta; same NMJ as in **b**). **c** Corresponding quantifications of immunostaining intensities in Tuba4a[KI] terminal motor axons (right; relative to Tuba4a[WT], left, which is set to 1) of Tubb3, normalized to neurofilament heavy polypeptide (WT: $n = 174$ axons, 10 mice; KI $n = 175$ axons, 10 mice, $p$-value = 0,01671) and of (**e**) polyE, normalized to tubulin beta-3 (WT: $n = 174$ axons, 10 mice;

KI $n = 175$ axons, 10 mice, $p$-value = 0,00771). **f** Percentage of polyinnervated NMJs in Tuba4a[WT] vs. Tuba4a[KI] littermate controls, based on tubulin beta-3 staining, at postnatal day (P) 6-7, 8-9, 10-11 and 12-13 (P6-7 WT $n = 4$, KI $n = 3$; P8-9 WT $n = 3$ KI $n = 3$; P10-11 WT $n = 3$, KI $n = 5$; P12-13 WT $n = 7$, KI $n = 4$ mice; ≥ 60 NMJs per animal, P6-7 $p$-value = 0.0310, P8-9 $p$-value = 0.005139, P10-11 $p$-value = 0.035714). Graphs: mean ± SEM in (**a**); 25–75% quantiles shown as a box with top and bottom black lines, mean as middle black line, and SEM as white lines (left); data representing single axons as dots (middle); half violin (right) in (**c**, **e**) or mean + SEM, with data representing animals as single dots (**f**). A two-sided Mann-Whitney test determined significance in (**c**, **e**, **f** P10-11) and a two-sided Student $t$ test in (**f** P6-7, P8-9): *, $P < 0.05$; **, $P < 0.01$. Scale bar, 5 μm in (**b**), applies also to (**d**). Source data are provided as a Source Data file.

CCP1&6[mnWT] controls (Supplementary Fig. 10). Importantly, the synapse elimination process in CCP1&6[mnKO] mice was accelerated, with fewer neuromuscular synapses being polyinnervated at several time points (Fig. 6m). In contrast, analysis of other aspects of neuromuscular development showed no differences in CCP1&6[mnKO] vs. CCP1&6[mnWT] (Supplementary Fig. 4g–i). Notably, the deglutamylase CCP1 is abundant in the CNS, similar to TTLL1 (Supplementary Fig. 6e–h). Thus, we sought to analyze hippocampal pruning and found a significantly accelerated pruning of the IPB in CCP1[KO] vs. wildtype controls at P14 (Fig. 7; Supplementary Fig. 11). These findings corroborate that the polyglutamylation writer and eraser enzymes bidirectionally regulate the developmental remodeling of axons and synapses.

### Spastin swiftly eliminates excessively polyglutamylated microtubules

The absence of excess polyglutamylation in young CCP1&6[mnKO] motor axons was unexpected, as CCPs remove glutamate residues from

microtubules; consequently, their genetic deletion should increase PolyE immunostaining. We explored two possible explanations for this observation: (i) excessive polyglutamylation might result in highly branched, arborized polyglutamyl chains that could evade PolyE-immunodetection due to steric hindrance[49] or (ii) instantaneous severing and subsequent destabilization of excessively polyglutamyl-decorated microtubules by enzymes such as spastin[30,31] might leave the polyglutamylated microtubule pool paradoxically depleted.

To test the former possibility, we performed anti-GT335 immunostainings, an antibody recognizing the branching point of glutamate chains on tubulin tails[26,50] independently of chain length. Confocal quantification indicated a similar staining intensity in CCP1&6[mnKO] mice compared to controls (Fig. 6n, o). This suggests that the absence of hyper-polyglutamylation in CCP1&6[mnKO] mice has a biological explanation, rather than a technical one. Next, we focused on a possible intersection of the effects of absent deglutamylase activity and spastin activation. To test this, we conducted neonatal (P3) injections of an adeno-associated virus (AAV9-hSyn-Cre), which expressed Cre

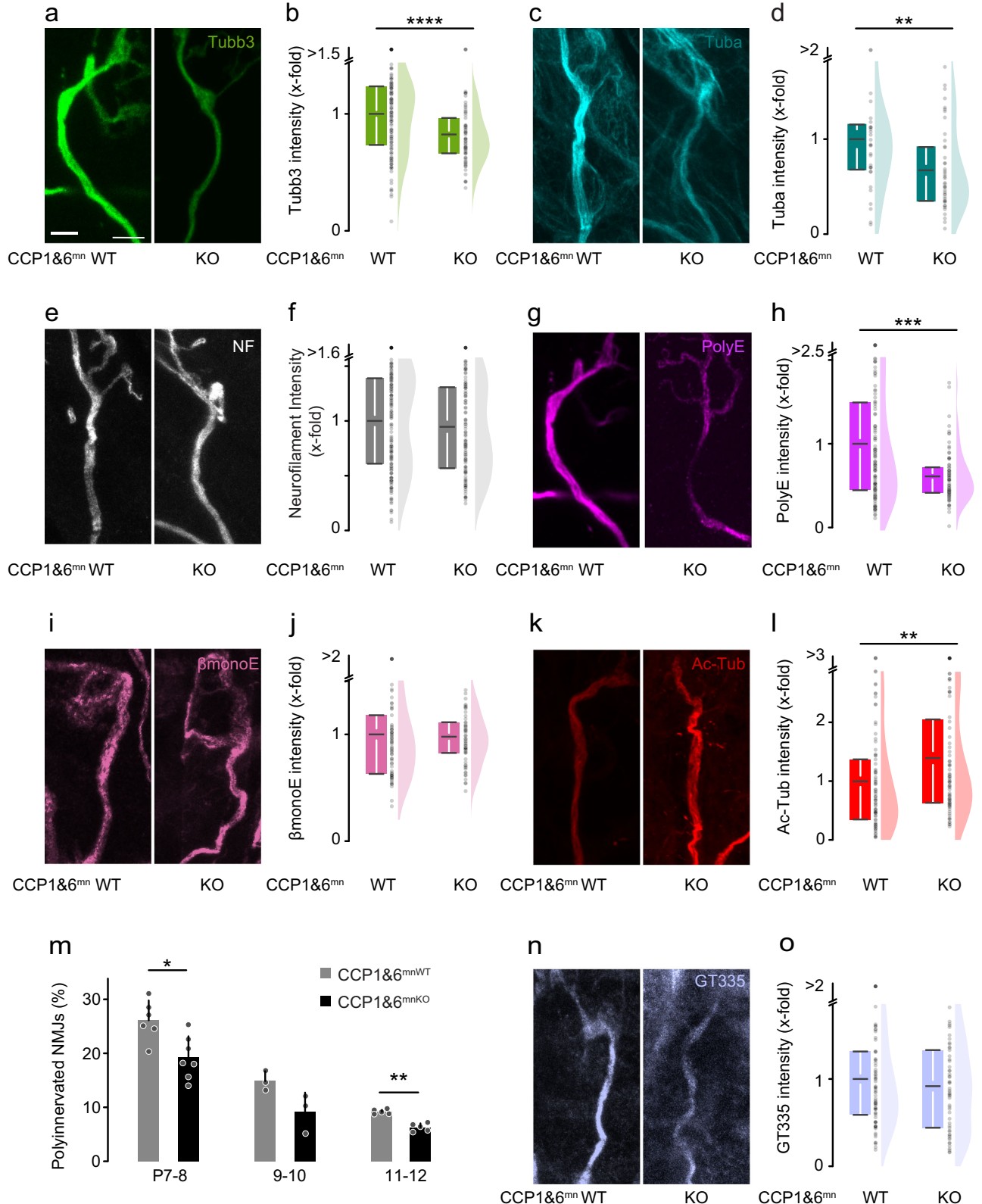

recombinase in neurons only, in conditional CCP1 and spastin knock-out mice that also carried a reporter allele (CCP1$^{flox/flox}$ X Spast$^{flox/flox}$ X TdTomato). The TdTomato reporter fluorescence was analyzed in motor axons at P9 to validate successful Cre-mediated excision (Fig. 8a). To circumvent background ambiguities and the lack of reliable in situ detection tools for endogenous spastin or deglutamylases, the tdTomato signal was indexed in quantiles based on fluorescent intensity. We compared the quantile with the highest intensity (tdTomato$^{high}$, which we considered most likely for CCP1 and spastin double-deletion) with the bottom quantile (tdTomato$^{low}$; likely non-recombined) for levels of PolyE immunostaining. This indeed revealed a 2-fold increase of PolyE in the tdTomato$^{high}$ compared to the tdTomato$^{low}$ axons (Fig. 8b, d), which was accompanied by added microtubule content (1.3-fold; Fig. 8b, c). Notably, spastin translation

**Fig. 6 | Genetic deletion of CCP1 and CCP6 in motor neurons accelerates pruning. a–k** Quantitative immunostainings for microtubule markers on triangularis sterni muscles at postnatal day (P) 9–12 from CCP1&6$^{mnKO}$ and CCP1&6$^{mnWT}$ littermate controls. Confocal stacks of terminal axons leading to NMJs depicting immunostainings for (**a**) tubulin beta-3 (Tubb3, green), (**c**) tubulin-alpha (Tuba, cyan), (**e**) neurofilament (NF, white), (**g**) polyglutamate chain staining (PolyE, magenta; same NMJ as in **a**), (**i**) monoglutamate on beta-tubulin staining (βmonoE, pink), (**k**) acetylated tubulin staining (Ac-Tub, red). **b–l** Corresponding quantifications of immunostaining intensities in CCP1&6$^{mnKO}$ terminal motor axons (relative to CCP1&6$^{mnWT}$ set to 1). **b** Tubb3, normalized to YFP (WT $n = 127$ axons, 4 mice; KO $n = 74$ axons, 3 mice, $p$-value = 5,0301E-5). **d** Tuba, normalized to NF (WT $n = 32$ axons, 2 mice; KO $n = 44$ axons, 4 mice; $p$-value = 0.00546). **f** NF (WT $n = 161$ axons, 7 mice, KO $n = 120$ axons, 5 mice). **h** polyE, normalized to Tubb3 (WT $n = 127$ axons, 4 mice; KO $n = 74$ axons, 3 mice, $p$-value = 7,84697E-4). **j** βmonoE intensity, normalized to Tubb3 (WT $n = 78$ axons, 4 mice; KO $n = 63$ axons, 3 mice). **l** ac-tub,

normalized to Tubb3 (WT $n = 85$ axons, 5 mice; KO $n = 80$ axons, 4 mice, $p$-value = 0.00233). **m** Percentage of polyinnervated neuromuscular junctions (NMJs) in CCP1&6$^{mnWT}$ vs. CCP1&6$^{mnKO}$ littermates crossbred to Thy1-YFP at P7-8, 9-10, and 11-12 (P7-8 WT $n = 6$, KO $n = 7$, $p$-value = 0.0221; P9-10 WT $n = 3$, KO $n = 3$; P11-12 WT $n = 5$, KO $n = 5$ mice, $p$-value = 0.0079; ≥125 NMJs per animal). **n** Confocal stack of a terminal axon leading to an NMJ depicting staining of branching points of glutamate side chains on beta- and alpha-tubulin (GT335, light blue). **o** Quantification of GT335 intensity in terminal motor axons, normalized to tubulin beta-3 (WT $n = 83$ axons, 3 mice; KO $n = 57$ axons, 3 mice). Graphs: 25−75% quantiles shown as a box with top and bottom black lines, mean as middle black line, and SEM as white lines (left); data representing single axons as dots (middle); half violin (right) in (**b, d, f, h, j, l, o**) or mean + SEM, with data representing animals as single dots (**m**). A two-sided Mann-Whitney test determined significance: **, $P < 0.01$; ***, $P < 0.001$; ****, $P < 0.0001$. Scale bar, $5\,\mu m$ in (**a**), applies also to (**c, e, g, i, k, n**). Source data are provided as a Source Data file.

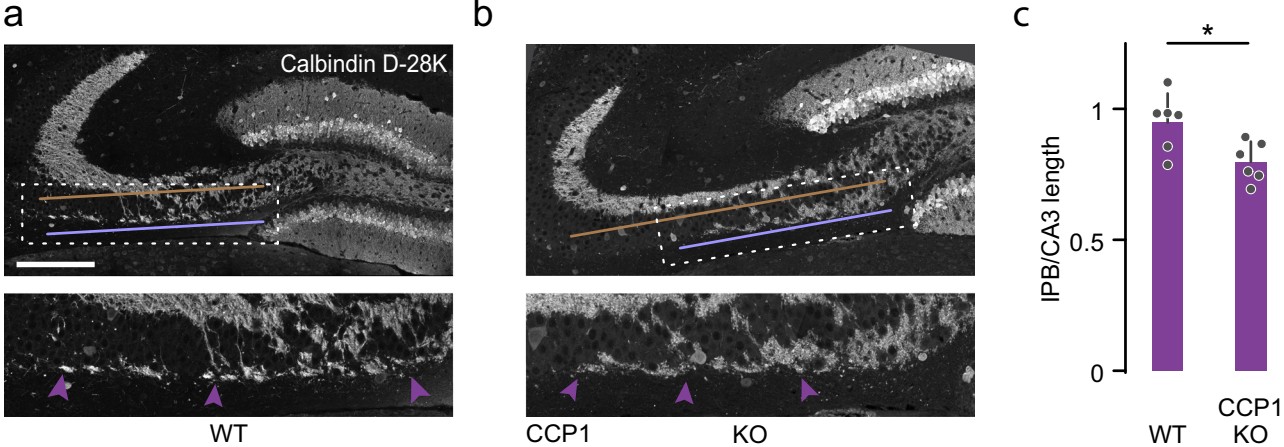

**Fig. 7 | CCP1$^{KO}$ leads to pruning defects in the CNS. a, b** Confocal stacks of coronal brain sections of P14 WT vs. CCP1$^{KO}$ mice immunostained for calbindin D-28k (white). The expansion of the dashed box shows IPB length at higher magnification (violet arrowheads); the light violet line indicates IPB length, and the brown line CA3 length is measured. **c** Quantification of IPB length normalized to CA3 length in

2-week-old WT vs. CCP1$^{KO}$ (WT: $n = 6$ hemispheres, 3 mice; KO: $n = 6$ hemispheres, 3 mice; $p$-value = 0.0411). Graphs: mean + SEM and data represent hemispheres as single dots. A two-sided Mann-Whitney test determined significance. *, $P < 0.05$. Scale bar, $100\,\mu m$ in (**a**), applies also to (**b**). Source data are provided as a Source Data file.

decreases from P5 onwards (which contrasts with other severing enzymes, such as Katnal1; Supplementary Fig. 4a and Supplementary Data 3). This suggests that spastin might be necessary during the developmental pruning phase−and, consistently, quantification of immunostainings in 6-week-old CCP1&6$^{mnKO}$ motor axons showed lower microtubule mass (0.4-fold; normalized to neurofilament; Supplementary Fig. 12b, c) accompanied by a drastic increase in PolyE intensity (normalized to Tubb3; 4.1-fold; Supplementary Fig. 12d, e), while neurofilament was similar in adult CCP1&6$^{mnKO}$ vs. CCP1&6$^{mnWT}$ littermate controls (Supplementary Fig. 12e, f). In summary, microtubule content and PTM composition in motor axons appear to be controlled by the interplay of tubulin glutamylases, deglutamylases, and specific microtubule severing enzymes.

### Microtubule PTM levels are under the control of neurotransmission

Neuronal activity is a primary driver of sculpting neural circuits, including at the NMJ[20], and has been demonstrated to facilitate tubulin polyglutamylation[51]. We previously showed that a block of neurotransmission by injection of α-bungarotoxin (α-BTX), which blocks postsynaptic acetylcholine receptors and stalls synapse elimination, leads to a reduction of tubulin beta-3 immunofluorescence in the presynaptic motor axon[52]. Moreover, polyglutamylation is lost in pruning axons with a small synaptic occupancy (Fig. 9a)[21]. To test if loss of tubulin beta-3 is mediated by activity-driven modulation of PTMs, we analyzed polyglutamylation in motor axons at P9 following an

injection of α-BTX into the thorax two days earlier (Fig. 9b). In axon branches that terminated in α-BTX-blocked synapses (BTX + ) polyglutamylation dropped significantly compared to control axons (0.7-fold; normalized to Tubb3; Fig. 9c, d and Supplementary Fig. 13a). This reduction following the block of neurotransmission is reminiscent of CCP1&6$^{mnKO}$ mice, where spastin appears to deplete polyglutamylated microtubules faster than glutamyl side chains can accumulate. To test this notion, we injected α-BTX into mice that globally lack spastin (Spast$^{KO}$) and accumulate polyglutamylation[21], enabling us to detect glutamylase activity upon block of neurotransmission. Indeed, PolyE immunostaining levels were increased (2.1-fold) in blocked axons (BTX + /Spast$^{KO}$) compared to branches innervating unaffected synapses (BTX-/Spast$^{KO}$; normalized to Tubb3; Fig. 9e, f and Supplementary Fig. 13b). Thus, our data suggest that neurotransmission governs the elongator TTLL glutamylase's enzymatic activity in the presynaptic axons. In summary, polyglutamylation indeed encodes activity-dependent signals that are subsequently read by spastin, which severs polyglutamylated microtubules and thus facilitates axon branch and synapse removal.

## Discussion

Neurons remodel extensively during development, with high temporal and spatial precision[20,53]. The fate of adjacent and otherwise indistinguishable neurites and synapses can starkly diverge−some are swiftly removed while others, often immediate neighbors, last for a lifetime[54]. The microtubule cytoskeleton plays a key role in such

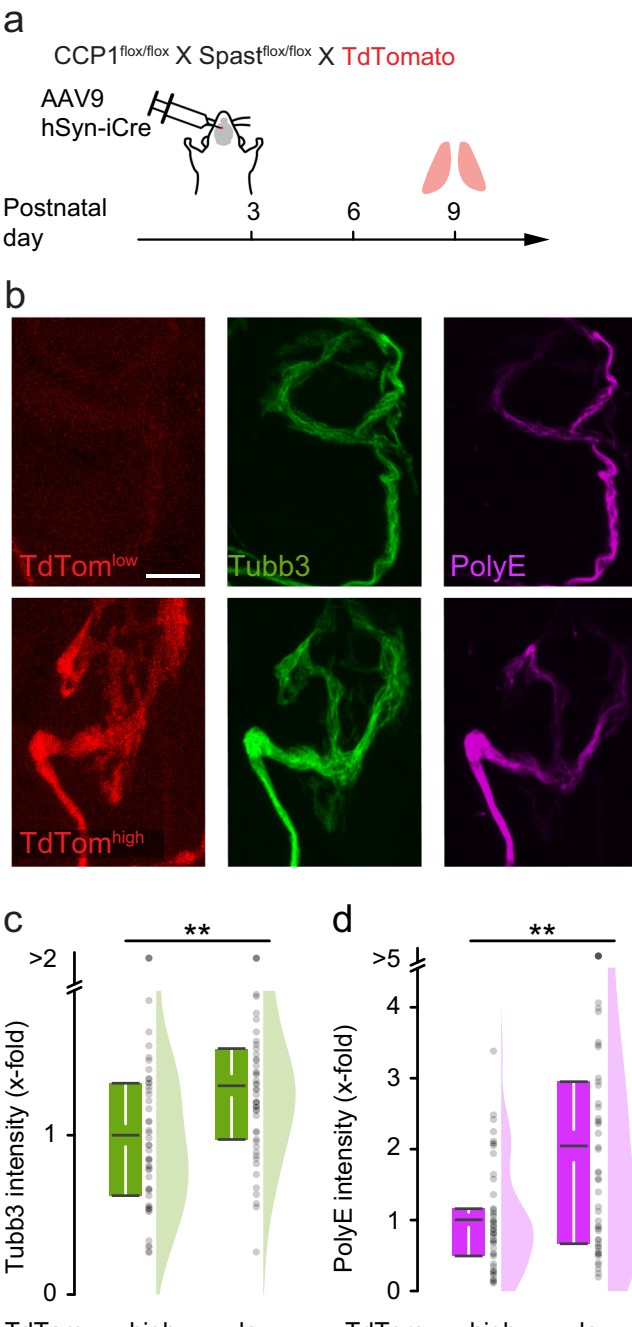

**Fig. 8 | Modulation of polyglutamylation by deletion of spastin and CCP1.**
**a** Schematic for acute deletion of CCP1 and spastin in a subset of motor neurons. Intraventricular injection on P3 into CCP1 flox/flox X Spast flox/flox X TdTomato reporter animals with viral vectors (AAV9-hSyn-iCre), followed by immunostaining for microtubule markers at P9. The expression of TdTomato (TdTom) served as a measure of CCP1 and Spastin deletion. Terminal motor axons were considered Cre recombined (TdTom^high) if their TdTomato intensity belonged to the 3rd quantile range and not recombined if they belonged to the 1st quantile range (TdTom^low). **b** Confocal stacks of NMJs depict TdTomato (red), Tubb3 (green), and PolyE (magenta) stainings. **c, d** Quantification in terminal motor axons of (**c**) Tubb3 (tom- 50 axons, 6 mice; tom + $n$ = 50 axons, 6 mice, $p$-value = 0,00126) and (**d**) PolyE, normalized to Tubb3 (tom- $n$ = 50 axons, 6 mice; tom + $n$ = 50 axons, 6 mice, $p$-value = 8,97637E-4). Graphs: 25–75% quantiles shown as a box with top and bottom black lines, mean as middle black line, and SEM as white lines (left); data representing single axons as dots (middle); half violin (right) in (**c, d**). A two-sided Mann-Whitney test determined significance: **, $P < 0.01$. Scale bar, 5 μm. Source data are provided as a Source Data file.

remodeling processes[17,34]. However, how cytoskeletal functions would be regulated in the required temporally- and spatially-controlled manner has remained unclear. Here, we show that enzymes involved in adding and removing polyglutamylation on tubulin determine the remodeling speed of axons and exert a rheostatic control of microtubule functions in vivo, as previously suggested in vitro[30]. This system sets the glutamylation level, which determines axon or synapse fate, e.g., via the microtubule destabilizing action of the reader severase spastin[21].

This concept is noteworthy from two vantage points: First, as a unique in vivo manifestation of the tubulin code concept, where PTMs convey specific local functionality to the cytoskeleton during neurodevelopment, and second, as a possible general cell biological mechanism of branch-specific axon or synapse dismantling. The tubulin code hypothesis suggests that PTMs modify microtubule function, including stability, dynamics, and interactions[4,55]. Polyglutamylation is a modification that is especially pronounced in neurons. By changing tubulin's charge characteristics, glutamylation can affect the recruitment of MAPs to microtubules[29,45]. Accordingly, polyglutamylation has been linked to transport deficits and consequent neurodegeneration[13,14,56,57]. However, physiological roles have been harder to decipher for this PTM. A large set of enzymes catalyze the addition and removal of glutamate residues, and the difficulty of probing them with sufficient spatial and temporal precision in the developing nervous system makes such investigations challenging. Our translatome-guided and cell-type-specific genetic approach now reveals TTLL1-mediated polyglutamylation of tubulin alpha-4A (Figs. 2 and 5) as an important step in neuronal pruning. It also corroborates the high enzymatic specificity of the tubulin polyglutamylation system in neural development[13], in concordance with the tubulin code concept. In contrast, the initiator glutamylase TTLL7, which catalyzes the addition of the first glutamate onto beta-tubulin[58], does not seem to be involved in remodeling (Fig. 3). Notably, tubulin alpha-4A—while not the most abundant tubulin isotype—is the only alpha-tubulin isotype that shows increasing expression across the neuromuscular remodeling period (Fig. 5a). This suggests that transcriptional regulation of tubulin genes and perhaps enzymes might be a global factor that is permissive for efficient polyglutamylation-driven remodeling. A further proposition to be tested based on increasing expression (Fig. 1e) is that TTLL5, an alternative initiator glutamylase for alpha-tubulin[59], might act upstream of TTLL1 during remodeling.

Our data includes further examples of the complex interactions among various players in this system, both on the transcriptional and post-translational levels. For instance, the motor neuron-specific deletion of deglutamylases (CCP1&6^mnKO) had different outcomes regarding microtubule mass and polyglutamylation levels in developmental remodeling vs. in adult motor axons (Supplementary Fig. 4b–e) —two time-windows that differed in spastin expression (Supplementary Fig. 4a). This drop in polyglutamylation could be reversed during remodeling by combined deletion of CCP1 and spastin (Fig. 7), suggesting that microtubule stability and apparent PTM levels are the results of the dynamic balance between eraser and reader activity. In this model, hyper-activated spastin typically removes polyglutamyl-decorated microtubules faster than this population can be replenished —leading to the observed paradoxical relative loss of polyglutamylated microtubules after ablation of the enzymes that generally remove this PTM. Moreover, the balance between the writer TTLL and the eraser CCP, resulting in a fine-grained level of local tubulin glutamylation, might regulate spastin's activity further since CCP1^KO rescues TTLL1^KO [14].

Polyglutamylation can profoundly vary even between adjacent axon branches that only differ in their subsequent fate[21]. Together with our data, this suggests that glutamate residues can be added to microtubules with high temporal and spatial precision. These locally modified microtubules can then, in turn, recruit and activate unique

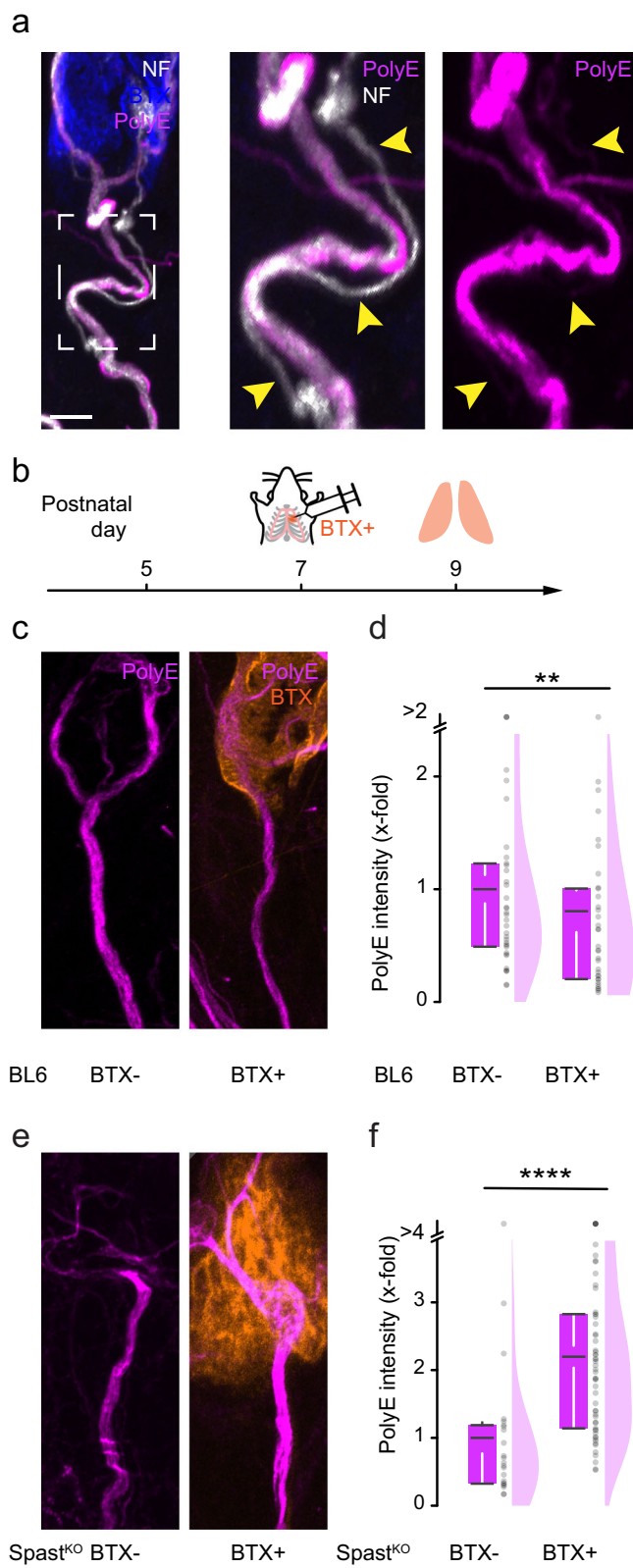

**Fig. 9 | Modulation of polyglutamylation by blockage of neurotransmission.**
**a** Confocal stack of wildtype (BL6) mice depicts a doubly innervated NMJ with a thicker 'winner' and a thinner 'loser' (yellow arrowheads) axon immunostained at P9 for PolyE (magenta), and α-BTX (blue; $n = 1$ mouse). The boxed area is shown at higher magnification on the right with single channels. **b** Schematic for α-BTX (conjugated to Alexa Fluor 594, orange) injections. Wildtype (BL6) and SpastKO mice were injected with α-BTX into the thoracic wall at postnatal day (P) 7. Triangularis sterni muscles were fixed and immunostained at P9 for microtubule markers. **c, e** Confocal stacks of (**c**) wildtype and (**e**) SpastKO NMJs depict non-injected (left) and BTX-injected motor axons (right; BTX, orange) stained for PolyE (magenta) at P9. **d, f** Quantification in terminal motor axons of polyE intensity normalized to Tubb3 of (**d**) wildtype (BTX-: $n = 37$ axons, 4 mice; BTX +: $n = 34$ axons, 4 mice; $p$-value = 0.04053) and (**f**) SpastKO mice (BTX-: $n = 23$ axons, 2 mice; BTX +: $n = 58$ axons, 3 mice; $p$-value 1,83216E-6). Graphs: 25–75% quantiles shown as a box with top and bottom black lines, mean as middle black line, and SEM as white lines (left); data representing single axons as dots (middle); half violin (right) in (**d, f**). A two-sided Mann-Whitney test determined significance: **, $P < 0.01$; ***, $P < 0.001$. Scale bar, 5 μm in (**a**), applies also to (**c**) and (**e**). Source data are provided as a Source Data file.

Overall, our work implies that polyglutamylation-based encoding can tag neurites for removal—a new model of selective axon removal that can be tested in other species and models.

Open questions remain regarding the events that occur upstream and downstream of the spastin-mediated microtubule destabilization. For instance, how does microtubule breakdown result in the characteristic but often divergent neuritic morphologies accompanying pruning? At the neuromuscular junction (NMJ), axonal breakdown involves the formation of axon bulbs, followed by piecemeal degradation, resembling apoptotic body shedding[32,53]. In contrast, mossy fibers in the infrapyramidal bundle and axons in remodeling neurons in the mushroom body seem to instead undergo fragmentation[53,60]. At the same time, spines are typically absorbed without a morphological trace[61]. Interestingly, many morphogenic processes in neurites are more commonly associated with actin dynamics rather than microtubule breakdown[62,63]. However, the switch between the growth and retraction of neurites involves an interplay of actin and microtubules[64]. Similarly, the balance between the longitudinally rigid co-axial microtubules vs. circumferentially-running actin striations might usually prevent axons from beading[65,66]. Thus, how the final steps of axonal disintegration that precede glial engulfment are executed by remodeling across the cytoskeleton, remain to be resolved.

Another key feature of some forms of neuronal remodeling is activity dependence[53,67]. It is not fully understood how altered activity patterns translate into morphogenic change. In our experiments, chronic α-BTX blockage of neuromuscular synapses (Fig. 8) changed the polyglutamylation levels in presynaptic motor axons. However, it remains elusive which factor in the enzymatic equilibrium that governs polyglutamylation levels is affected and how the enzymes that mediate or interpret polyglutamylation are regulated by the transcellular signaling that underlies synapse elimination[64]. For example, spastin phosphorylation might regulate its binding to microtubules or protein stability, although it is unclear how this would impact severing[68]. For TTLLs and CCPs, information about how their enzymatic activity is regulated is even sparser[40,69]. Indeed, to our knowledge, thus far, no link has been made between regulatory mechanisms that control polyglutamylation and the signaling pathways that are presumed to execute neurite remodeling[53]. Our prior work provides a possible feedback mechanism by which the relevant enzymes might be supplied by microtubule-dependent transport, which we have shown to stall early in remodeling[21] and influence other signaling events, such as myelination[52]. Thus, changes in transport could alter the local equilibrium between the addition and the removal of glutamate residues, with spastin acting as a local, branch-specific detector.

sets of MAPs, including tau, kinesin motors, and severing enzymes, such as spastin[31,29,30,45]. Our data also reveal that changing one PTM (e.g., via genetic manipulation of TTLL1) can percolate within the cytoskeleton to affect other PTMs, such as acetylation[40]. Thus, the interplay between various PTMs results in a complex pattern of specialized microtubule subsystems within individual axon branches that determines pruning via microtubule stability and function regulation.

Indeed, our work strengthens the long-sought link between developmental neurite remodeling and axon degeneration[53]. Many of the enzymes of the remodeling mechanism that we describe here are linked to neurodegenerative disease: Spastin to hereditary spastic paraplegia[9], CCP1 to infantile-onset neurodegeneration[70], and Tuba4a to motor neuron disease[45]. A unifying mechanism could involve alterations in microtubule dynamics, followed by altered organelle and enzyme transport, which again could feed back to cytoskeletal stability[71]. Thus, developmental neurite remodeling could be instructive for developing strategies to target microtubules in settings of axon degeneration by focusing drug development on polyglutamylation.

## Methods

### Mouse lines, husbandry, and genotyping
All animal experiments conform to the German federal law. The protocols were examined by the Government of Upper Bavaria and approved by the ethics committee according to §15 German Animal Welfare Act, in agreement with the government of Upper Bavaria. In all experiments, mice of both sexes were included. Experimental animals were kept together with littermates. The animals were housed in groups in individually ventilated cages (IVC) on dust-free wood chips with red plastic houses, paper rolls, gnawing sticks, wood wool, and fed with standard mouse chow. Ambient temperature in the cages was kept between 20 and 24 °C, humidity between 35% and 55%, and a light/dark cycle of 12 h/12 h.

Motor neuron knockout-specific mice were generated by crossbreeding conditional knockout mice to ChAT-IRES-Cre[35] mice, expressing Cre-recombinase under the ChAT promoter (Jackson #6410). Ribotagging of motor neurons was conducted in ChAT-IRES-Cre crossbred to homozygous Rpl22$^{HA}$ [36] mice (RiboTag$^{flox/flox}$; Jackson, #11029) and conditional Spast$^{flox/flox}$ [21] (Fig. 1 and Supplementary Fig. 1 and 12). Glutamylases (TTLL1 or TTLL7) or deglutamylases (CCP1 and CCP6) were deleted in motor neurons by crossbreeding conditional mutants of CCP1[72] &6[14], TTLL1[14] or TTLL7[13] mice (from Janke lab, Institut Curie, Orsay) to ChAT-IRES-Cre[35] animals. All animals were CCP1&6, TTLL1, or TTLL7 homozygous. Experimental animals – CCP1&6$^{mnKO}$, TTLL1$^{mnKO}$ or TTLL7$^{mnKO}$ – were ChAT-IRES-Cre-positive, whereas littermate controls were ChAT-IRES-Cre-negative, named hereafter CCP1&6$^{mnWT}$, TTLL1$^{mnWT}$ or TTLL7$^{mnWT}$. Thy1-YFP[73] transgenic mice (cytoplasmic YFP in all motor neurons, Jackson #3709) were used to assess pruning speed in cross-breeding to CCP1&6$^{mnKO}$, TTLL1$^{mnKO}$, TTLL7$^{mnKO,}$ compared to their littermates CCP1&6$^{mnWT}$, TTLL1$^{mnWT}$, TTLL7$^{mnWT}$. Microtubule dynamics were visualized and analyzed by Thy1-EB3-YFP[41] transgenic animals crossed to either CCP1&6$^{mnKO}$, TTLL1$^{mnKO}$, or TTLL7$^{mnKO}$; CCP1&6$^{mnWT}$, TTLL1$^{mnWT}$, or TTLL7$^{mnWT}$ were littermate controls. Conditional knockout of CCP1[72] and Spastin[21] in motor neurons, monitored by ROSA-CAG-TdTomato reporter[74] (Ai14; Jackson; #7914; CCP1$^{flox/flox}$ X Spast$^{flox/flox}$ X TdTomato) was generated by injection of viral vectors encoding Cre-recombinase. Block of neurotransmission was conducted and analyzed in constitutive spastin knockout[21] (Spast$^{KO}$) mice (homozygous Spast$^{KO}$ vs controls Spast$^{WT}$) and C57BL/6 N (Charles River, Strain Code 027) controls injected with α-BTX. Spine density measurements and hippocampal pruning analysis were performed on TTLL1 constitutive knock-out[14] at 3 and 8 weeks of age (P21 and P56; named here TTLL1$^{KO}$; from Janke lab, Institut Curie, Orsay, France), backcrossed to CD-1 (Charles River, Strain Code 022) for three generations. TTLL1$^{KO}$ and TTLL1$^{WT}$ littermates were generated by crossbreeding conditional TTLL1 to Pgk1-cre[75] (Jackson, #020811; TTLL1$^{KO/WT}$ X TTLL1$^{KO/WT}$). CCP1 hippocampal pruning analysis was performed on constitutive knock-out[72] (named here CCP1$^{KO}$; from Janke lab, Institut Curie, Orsay, France) at P14, generated by crossbreeding conditional CCP1 to CMV-cre expressing mice vs. P14 wildtype controls. Cytoskeletal analysis and pruning speed were analyzed in homozygous tubulin alpha-4A knock-in[45] (Tuba4a$^{KI/KI}$) mice (from

Kneussel lab, ZMNH, Hamburg, Germany), and negative, thus wildtype, littermates (Tuba4a$^{WT/WT}$) served as controls.

Genotyping was performed as described[21,52] (for the Tuba4a$^{KI}$, with variations as published[45]). Briefly, genomic DNA was extracted from biopsies using one-step lysis (lysis buffer in mM: 67 Tris, pH 8.8, 16.6 (NH$_4$)$_2$SO$_4$, 6.5 MgCl$_2$, 5 β-mercaptoethanol, 10% Triton-X-100, and 50 µg/ml Proteinase K; incubation at 55 °C for 5 h, followed by inactivation step 5 min at 95 °C). PCR was performed with GoTaq Green Master Mix (Promega; #M7121) following a standard protocol, and then DNA was separated on a 1.5–2% agarose gel. Primer sequences are listed in Supplementary Data 4.

### Ribosomal pull-down and RNA sequencing
Spinal cords were dissected from ChAT-IRES-Cre X Rpl22$^{HA}$ X Spast$^{flox}$ mice (see Mouse lines, husbandry, and genotyping above). The samples were kept at −80 °C until the pull down of labeled ribosomes described previously[36] (see also http://depts.washington.edu/mcklab/RiboTag.html). Spinal cords were homogenized (buffer in RNase free water in mM: 50 TrisCl, pH7.5, 100 KCl (Sigma #P9541), 12 MgCl$_2$ (Sigma #63068), 1% NP-40 (Roche, #11332473001), 1 X dithiothreitol (DTT, Sigma #646563), 1 X protease inhibitors (Sigma #11697498001), 1 mg/ml heparin (Sigma #H3393), 100 µg/ml cycloheximide (Sigma #C7698), RNAseOUT (Thermofisher #10777019). As previously described, the mRNA-ribosome complex was precipitated using a polyclonal HA-antibody (Sigma #H6908) and Dynabeads Protein G (Life Technologies #10004D). Ribosome-bound mRNA was isolated with RNeasy Plus Micro Kit (Qiagen #74034) per manufacturer instructions (not precipitated fraction was used as input spinal cord control). Prior sequencing, RNA quantity and integrity (RIN > 8.5) were controlled with a Bioanalyzer (Agilent RNA 6000 Nano), and rRNA was depleted. Motor neuron mRNA was sequenced in pair-end using an Illumina HiSeq4000 Kit at a depth of ~40 million reads per sample (Institute of Neurogenomics, Juliane Winkelmann, Helmholtz Munich, Germany). The raw sequencing data (fastq files) were aligned to the mm9 mouse genome, and read counts were extracted with HTSeq-count software with the option intersectionStrict. Lowly expressed RNAs (<10 read counts were excluded, and differential gene expression analysis was performed with the DESeq2 software package (Bioconductor.org)[76]. For each gene, a Wald test was applied to assess differences in expression between conditions. The tests were two-sided, and p-values were adjusted for multiple comparisons using the Benjamini-Hochberg method to control the false discovery rate (FDR). Graphs were generated using ggplot2 from custom R statistical software scripts[77].

### Reverse transcription (RT)-qPCR
All the following steps were conducted on ice. cDNA was obtained according to the manufacturer's instructions. Briefly, 100 ng ribo-tagged motor neurons mRNA, 100 µM random hexamer primers (Roche # 11034731001), and 1 µl RNAsin Plus Inhibitor (Promega #N2615) were dissolved with water in 14 µl total volume. The mixture was incubated for 5 min at 70 °C and 10 min on ice. 5 µl of M-MLV Reverse Transcriptase Buffer (Promega #M531A), 10 mM NTPs mixture, and 200 µ of MMLV (Promega #M170A) were added, and the mixture was incubated for 1.5 h at 37 °C. cDNA was purified with a QIAEX II Gel Extraction Kit (Qiagen #20021) as follows: The obtained cDNA was mixed with 2 µl QiaEX-II suspension (Silica-Matrix) and 80 µl QX-I buffer, incubated for 20 min at 25 °C at 1000 rpm and centrifuged at 13,000 x g for 2 min. The supernatant was discarded, and the matrix was washed twice with 90 µl PE buffer and centrifuged at 13,000 x g for 2 min. The supernatant was removed, and the pellet dried for 10 min with the lid open at 300 rpm, shaking at 25 °C. The cDNA was eluted with 20 µl elution buffer (1 mM Tris, pH 8.5) and incubated at 25 °C with 1000 rpm for homogenization. The supernatant containing the clean RNA was collected after centrifugation at 13,000 x g for 2 min.

All qPCR reactions were performed with a LightCyler 1.3 Real-Time PCR system (Roche S/N: 140 6143). The MgCl$_2$ concentration (1–3 mM) and the annealing temperature were optimized for each primer pair and confirmed through primer efficiency curves 40. Primer sequences were (5′ to 3′): Chat: TTCTAGCTGTGAGGAGGTGC and CCCAAACCGCTTCACAATGG, GFAP: TCGCACTCAATACGAGGCAG and TTGGCGGCGATAGTCGTTAG. For each reaction, a final concentration of 1 ng/µl cDNA was mixed with 2 µl FastStart DNA Master SYBR Green I (Roche #03 003 230 001), 0.5 µM forward and reverse primer, 1–3 mM MgCl$_2$ and water to a total volume of 20 µl. Samples were run at least in triplicate. Controls omitting either template or reverse transcriptase were included for each primer pair.

## Immunofluorescence

Following dissection of the skin and pectoral muscles overlying the thorax, the whole thorax was fixed in 4% PFA for 1 h in 0.1 M phosphate buffer (PB) on ice. The triangularis muscle on the inner side of the thoracic wall was dissected and extracted[21,34,39,52,78]. Before applying primary antibodies, muscles were incubated in 5% CHAPS (Carl Roth #75621-03-3) in 0.1 M PB for 1 h at 37 °C. Primary antibodies (listed below) were diluted in blocking solution (5% BSA, Sigma #A7030; 0.5% Triton X-100 (Sigma) in 0.1 M PB; or 3% BSA, 0.5% Triton X-100 and 12% NGS, Abcam #ab7481 in 0.1 M PB) and incubated at 4 °C overnight for postnatal muscles, 3 days when adult tissue or anti-acetylated tubulin antibody was used. The following primary antibodies were used in this study: anti-tubulin beta-3 conjugated to Alexa Fluor 488 (BioLegend #657404, 1:200, clone AA10), anti-tubulin beta-3 conjugated Alexa Fluor 555 (BD PharMingen #560339, 1:200, clone TUJ1), anti-tubulin beta-3 conjugated Alexa Fluor 647 (BioLegend #657405, 1:200, clone AA10), anti-alpha-tubulin conjugated to Alexa Fluor 647 (Abcam #ab195889, 1:250, clone EP1332Y), anti-Polyglutamylation Modification (Adipogen #AG-20B-0020, 1: 200, clone GT335), anti-Polyglutamate chain (Adipogen #AG-25B-0030, 1: 1000, polyclonal), anti-alpha tubulin (acetyl K40, Abcam #ab24610, 1: 1000, clone 6-11B-1), anti-neurofilament heavy polypetide (Abcam #ab4680, 1:500, polyclonal), anti-βmonoE (Adipogen # AG-25B-0039, 1:500, polyclonal). Muscles were washed in 0.1 M PB, incubated for 1 h at room temperature with corresponding secondary antibodies coupled to Alexa Fluor 488, Alexa Fluor 594, or Alexa Fluor 647 (anti-rabbit: ThermoFisher #A-11070, 1:1000, polyclonal; ThermoFisher #A-11072, 1:1000, polyclonal; ThermoFisher #A-21246, 1:1000, polyclonal; ThermoFisher #A-32790, 1:1000, polyclonal; anti-mouse: ThermoFisher #A-11005, 1:1000, polyclonal; anti-chicken: ThermoFisher #A-11042, 1:1000, polyclonal; ThermoFisher #A-21449, 1:1000, polyclonal; anti-goat: ThermoFisher #A-11058, 1:1000, polyclonal) and washed again in 0.1 M PB. Some muscles were counterstained (Supplementary Fig. 4) with phalloidin Alexa Fluor 594 (ThermoFisher #A12381, 1:500) or α-BTX (ThermoFisher #B13423, ThermoFisher #B13422, ThermoFisher #B35450). Muscles or sections were mounted in Vectashield (Vector Laboratories # H-1000-10) or Prolong-Glass (Thermo Fisher #P36984). Image stacks were recorded at a confocal microscope (Olympus; FV1000 or FV3000) equipped with ×20/0.8 NA and ×60/1.42 NA oil-immersion objectives (Olympus). To immunostain spinal cord sections (Fig. 1), mice were transcardially perfused with PBS and ice-cold 4% PFA. Spinal cords were dissected and post-fixed overnight with 4 % PFA. 60 µm-thick vibratome sections were then immunostained with antibodies against choline acetyltransferase (ChAT; anti-ChAT, Novus Biologicals #NBP1-30052, 1:10) and hemagglutinin (HA; anti-HA, Sigma-Aldrich #H6908, 1:50, polyclonal), for 2 days overnight. Spinal cord sections were washed in 1x PBS and incubated for 1 h at room temperature with corresponding secondary antibodies (see above). The antibodies for spinal cord staining were diluted in blocking solution prepared with 5% normal donkey serum (Millipore #S30-M) and 0.5% Triton-X in 1x PBS.

Immunostaining of brain sections (Fig. 4) was carried out as described previously[43]. Briefly, mice were transcardially perfused with PBS and ice-cold 4% PFA in PBS, and brains were post-fixed in 4% PFA in PBS overnight, embedded in paraffin blocks, and cut at 7 µm-thick coronal sections. Prior to staining, antigen retrieval (citrate-based antigen unmasking solution, Vector Laboratories) and blocking (1% BSA, 0.2% Tween 20) were performed. For IPB, visualization sections were incubated overnight at 4 °C with anti-Calbindin D-28K antibody (Swant #CB-38a, 1:300, polyclonal) diluted in blocking solution. Subsequently, samples were washed three times with PBS with 0.2% Tween20 and incubated for 2 h at room temperature with fluorescent secondary antibody coupled to Alexa Fluor 594 or Alexa Fluor 488 (ThermoFisher #A-11037, 1:500, polyclonal; ThermoFisher #A-11008, 1:500, polyclonal) in blocking solution, washed three times with PBS with 0.2% Tween 20 and embedded in Mowiol (Sigma #81381) with 1 µg/ml Hoechst (Sigma #14533). Images were recorded using a confocal microscope (Leica SP8 and Leica Stellaris).

## DiOlistics labeling of hippocampal granule cells and dendritic spine density analysis

For visualization of dendritic spines in TTLL1$^{KO}$ brains, a DiOlistic approach was used as described previously[43]. In brief, mice were perfused with 20 ml of 4% PFA/PBS, and brains were isolated and post-fixed in 4% PFA/PBS for 30 min. Then, brains were washed in PBS for 30 min and incubated in 15% sucrose for 30 min, followed by 30 min in 30% sucrose. 250 µm thick coronal slices were cut in vibratome, washed in PBS, and incubated for 5 min in 15% sucrose and 30% sucrose. The solution was removed, and DiI was introduced into the brain slices by DiI-labeled tungsten particles (prepared as previously described[43]) using a helium-powered Gene Gun system (Bio-Rad) at 120 psi pressure. After the labeling, slices were washed in PBS to remove residual tungsten particles and kept in PBS for 30 min in the dark to let the dye diffuse. Slices were mounted onto a glass slice in 0.5 % n-propyl gallate/ 90% glycerol/PBS (NPG) and the next day imaged using a confocal microscope (Leica Stellaris). The dendritic spine density was analyzed in 3 different TTLL1$^{WT}$ and TTLL1$^{KO}$ brains (at least 10 labeled granule cell dendrites were analyzed per brain).

## Virus production

AAV9-hSyn-iCre was produced as outlined previously[79]. In brief, HEK293-T cells were seeded in 10-tray Cell Factories (Thermo Fisher Scientific) 24 h before transfection, allowing the cells to reach 80-90% confluence. Subsequently, 420 µg of the vector plasmid dsAAV-hSyn-iCre and 1.5 mg of the helper plasmid pDP9rs, generously provided by Roger Hajjar (Phospholamban Foundation, Amsterdam, The Netherlands), were introduced into the cells by polyethyleneimine-mediated transfection (Polysciences, 24765 Warrington, PA). Cells were harvested after 72 h, followed by lysis and benzonase treatment. To purify the AAVs, ultracentrifugation on an iodixanol density gradient (Progen, OptiPrep) was applied. Buffer exchange from iodixanol to Ringer's lactate buffer was then carried out using Vivaspin 20 columns (Sartorius, VS2042, Göttingen, Germany). The virus content of two 10-tray Cell Factories was pooled and concentrated to obtain high virus titers. Real-time qPCR with SYBR Green Master Mix (Roche) was performed to evaluate AAV9 titers.

## Neonatal AAV9 or α-BTX injections

Viral vectors were injected into neonatal pups according to previously published protocols[21,52,80]. In short, P3 pups were briefly anesthetized with isofluorane (Abbott) and injected with 3 µl of AAV9-hSyn-iCre (titer $1 \times 10^{13}$ to $2 \times 10^{14}$) into the right lateral ventricle using a nano-liter injector (World Precision Instruments; Micro4 MicroSyringe Pump Controller connected with Nanoliter 2000) attached to a fine glass pipette (Drummond; 3.5", #3-000-203-G/X) at a rate of 30 nl/s. The injection was guided by ultrasound (Visualsonics, Vevo® 2100).

0.05% (wt/vol) trypan blue was added to the viral solution to visualize the filling of the injected ventricles. Whole litters were injected, and pups were allowed to recover on a heating mat before the litter was returned to their mother in the home cage and sacrificed on P9 for experiments.

For analysis, Cre-mediated deletion in conditional CCP1 and Spast knockout mice (CCP1[flox/flox 72] X Spast[flox/flox 21]) was verified by a robust expression of the TdTomato reporter allele[74] (homozygous) in motor neurons of the triangularis sterni muscle. According to TdTomato fluorescence intensity in axons, recombination was assumed when fluorescence was in the upper quantile and negative when they belonged to the lowest quantile. All other axons were excluded from the analysis.

To block neurotransmission in the triangularis sterni muscle of C57BL/6 N (Charles River, Strain Code 027) or Spast[KO 21] mice, 1 μl of 50 μg/μl α-BTX conjugated to Alexa Fluor 594 (ThermoFisher #B13423) was injected unilaterally with a needle in the thorax on P7, as previously described[52]. In some cases, a post-hoc stain of the triangularis sterni muscle with anti-neurofilament (see above) or α-BTX Alexa Fluor 488 or 647 (ThermoFisher #B13422, ThermoFisher #B35450) further verified the degree of labeling with injected α-BTX conjugated Alexa Fluor 594 and the absence of denervation (>100 NMJs per mouse, $n = 3$ mice). Immunostaining and quantification were conducted as described above with anti-tubulin beta-3 and anti-polyE antibodies.

### Live imaging of EB3 comet densities in nerve-muscle explants

EB3 comet imaging was performed on acute nerve-muscle explants as described[21,41]. In brief, the thorax of euthanized mice was obtained by removing the skin over the rib cage, severing the ribs close to the spinal column, and dissecting the diaphragm. Further dissections were done under oxygenated, ice-cold Ringer's solution (in mM: 125 NaCl, 2.5 KCl, 1.25 NaH$_2$PO4, 26 NaHCO$_3$, 2 CaCl$_2$, 1 MgCl$_2$, and 20 glucose, oxygenated with 95% O$_2$/ 5% CO$_2$) to remove thymus, pleura, lung and pectoral muscles over the rib cage. The explant was fixed with insect pins (Fine Science Tools; 26001–25, 0.25 mm) on a Sylgard-coated 3.5 cm petri dish, with the inside of the thorax facing the objective. Imaging was performed under continuous and steady perfusion with warmed oxygenated Ringer's solution and kept at physiological temperatures 33–36 °C by a heated stage connected to an automatic temperature controller (Warner Instruments; TC-344C). Total imaging time on explants did not exceed 2 h. Live imaging was carried out with an epifluorescence microscope (Olympus BX51WI) equipped with × 20/0.5 NA and × 100/1.0 NA water-immersion objectives, an automated filter wheel (Sutter Instruments; Lambda 10–3), a charge-coupled device camera (Visitron Systems; CoolSnap HQ2), controlled by μManager version 1.4[81]. 200 frames were acquired per movie at a frequency of 0.5 Hz and an exposure time of 500 ms using a YFP filter set (F36-528; AHF Analysentechnik).

### Data analysis

ImageJ/Fiji[82] (http://fiji.sc) was used to determine pruning speed by counting the number of innervating terminal branches ending on each α-BTX-stained NMJ. Three background-subtracted areas (ROIs, measured for mean gray value) were averaged per single axon to quantify immunostainings. Each ROI was obtained from a single optical section, as described[21,52]. IPB length was quantified using the ratio of IPB length to the length of the CA3 as described previously[83]. To quantify EB3 comet parameters, out-of-focus frames were deleted manually from movies and aligned with the TurboReg plugin with the parameter Rigid-body and Quality set on Accurate, and comet trajectories were manually analyzed using the MTrackJ plugin (developed by E. Meijering, Biomedical Imaging Group, Erasmus Medical Center, Rotterdam). Only EB3 comets that appeared for at least three consecutive frames were considered. Neurolucida software (MBF Bioscience) was used to perform dendritic spine segmentation.

For image representation, maximum intensity projections were generated from confocal image stacks with ImageJ/Fiji[82] and then further processed in Adobe Photoshop.

### Statistics & reproducibility

Statistical analysis and visualization were performed using R (r-project.org)[77], DESeq2 software package (Bioconductor.org)[76], OriginLab software (origin.lab), or GraphPad Prism version 10 for Windows. A two-tailed Student's $t$ test was performed to determine once the data passed the normality test. Otherwise, the Mann-Whitney test was used for non-parametric data. $P < 0.05$, indicated as *, was considered significant. $P < 0.01$ is **, $P < 0.001$ is ***, and $P < 0.0001$ is ****. Graphs for axons: 25%– 75% quantiles as a box with top and bottom black lines + mean as middle black line + SEM as white lines (left) + data representing single axons as single dots (middle). Half violin (right) depicts data distribution and is cut at 90% level of the dominant violin and at 0. Graphs for animals: mean + SEM, with data representing single animals.

No statistical method was used to predetermine the sample size. No data were excluded from the analyses. All analyses were performed with the investigators blinded to the genotypes and treatment during experiments and outcome assessment. Low-quality RNA (RIN < 8.5) was not sequenced. Lowly expressed RNAs (<10 read counts) were excluded from differential gene expression analysis.

### Reporting summary

Further information on research design is available in the Nature Portfolio Reporting Summary linked to this article.

### Data availability

The immunostaining data generated in this study are provided in the Source Data file. The RNA-seq data generated in this study have been deposited in the Gene Expression Omnibus (GEO) database under accession number GSE296782. Source data are provided in this paper.

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

## Acknowledgements

We thank Emily Spaulding and Rob Burgess (The Jackson Laboratory, Bar Harbor, ME, USA) for providing the RiboTag protocol; Julien Gagneur (Technical University of Munich, Computational Molecular Medicine, Munich, Germany) for advice on RNA sequencing analysis; Leanne Godinho for critically commenting on an earlier version of the manuscript and Adrian Marti Pastor for software advice. For excellent technical assistance, we thank Kristina Wullimann, Sabine Brummer, and Yvonne Hufnagel; for animal husbandry, we acknowledge Daniela Steinmetz, Manuela Budak and Nebahat Budak. We also thank Kristine Kellermann for veterinarian consultancy, as well as Monika Schetterer and Sebastian Berger for administrative and data management support. TM and MSB were funded by the *German Research Foundation* (DFG) Excellence Cluster SyNergy (EXC 2145 – ID 390857198). MSB is the recipient of a DFG research grant (LE 4610/1-1 – ID 450131873) and is supported by the DGM foundation. TM was further supported by the European Research Council under the European Union's Seventh Framework Program (grant no. FP/2007-2013; ERC Grant Agreement no.: 616791), the German Center for Neurodegenerative Disease (DZNE), by DFG Mi 694/9-1 (FG Immunostroke 428663564) and by the DFG TRR 274/1 2020 (projects C02 and B03; ID 408885537). S.E. was supported by the DFG (ID 403584255–TRR267) and the Federal Ministry of Education and Research (BMBF) in the framework of the Cluster4future program (CNATM - Cluster for Nucleic Acid Therapeutics Munich). MK was supported by the DFG research grant KN556/18-1 (ID 523862973). M.B. was supported by the Czech Science Foundation grant 21-24571S. MR was supported by the project National Institute for Neurology Research (Program EXCELES, ID Project No. LX22NPO5107), by the European Union–Next Generation EU, and by the Grant Agency of Charles University, grant GAUK 275423. CJ is supported by the program Investissements d'Avenir launched by the French Government and implemented by ANR with the references ANR-11-LBX-0038 and ANR-10-IDEX-0001-02 PSL, by the Institut Curie, the French National Research Agency (ANR) awards ANR-17-CE13-0021 and ANR-20-CE13-0011, and the Fondation pour la Recherche Medicale (FRM) grants DEQ20170336756 and FRM MND2020030. MM obtained funding from the Fondation Vaincre Alzheimer grant FR-16055p and the France Alzheimer grant 2023.

## Author contributions

Conceptualization A.G., A.Z., C.J., T.M. and M.S.B.; data curation A.G. and A.Z.; methodology M.B., T.M. and M.S.B.; funding acquisition T.M. and M.S.B.; investigation A.G., A.Z., M.R., M.W. and A.I.U.; resources K.A.Z., M.M., S.C., T.J.H., S.E., M.K., C.J., T.M. and M.S.B.; supervision S.E., M.K., M.B., C.J., T.M. and M.S.B.; visualization A.G., A.Z. and M.S.B.; writing – original draft A.G., T.M. and M.S.B.; writing – review & editing all authors.

## Funding

## Competing interests

The authors declare no competing interests.
