## [Transparent Peer Review file · Nature Communications]

Polyglutamylation of microtubules drives neuronal remodeling

Corresponding Author: Dr Monika Brill

Version 0:

Reviewer comments:

Reviewer #1

(Remarks to the Author)

In this study the authors address the microtubule (MT) posttranslational modification (PTM) polyglutamination in the context of motor axon and central nervous system axon pruning. They employ mutants in genes encoding seeder/initiator and elongator glutaminases (TTLL7 and TTLL1, respectively), and also deglutaminases (CCP1 and 6) that they show here via transcriptomic profiling are expressed in motor neurons over developmental timepoints that encompass motor axon pruning. The central results show that loss of TTLL function can delay neuromuscular synapse elimination and axon pruning, whereas loss of CCP function has the opposite effect. Additional work investigates the role played by the MT severing protein spastin in regulating MT polyglutamination and motor axon pruning, and also an initial assessment of neural activity's role in MT polyglutamination levels. Overall this is an interesting study that addresses a significant issue and demonstrates a novel role for MT PTMs in remodeling and axon pruning in at the NMJ, and in less detail in pruning of the hippocampal infrapyramidal tract (IFT). Attention to the points raised below, mostly minor, would strengthen this study.

Weaknesses

- data normalization: there are multiple examples in immunohistochemistry experiments where authors do not show fluorescence channel for signal normalization (e.g. neurofilaments). Authors also normalize polyglutamylation signals to β -tubulin, but this modification is mostly prominent on α -tubulins.

Major comments

In Fig. 1d-f, TTLL1 is seen to be high throughout the development (P5 – P14), both TTLL7 and TTLL11 are observed to be increasing, and also CCP1 is also over postnatal timepoints. One might expect that if TTLLs are involved in pruning, their levels might increase without a concomitant increase in CCP. The authors show that in the TTLL1mnKO (Fig. 2j), there is already a high level of poly-innervated NMJs at P8-9. In line with this observation, the CCP1/6 cKO has fewer poly-innervated NMJs already at P7-8 (Fig. 6i). Together, this suggests that TTLL/CCPs participate significantly in the initial formation of the NMJ. Support for the notion that "Genetic ablation of TTLL1 delays motor axon remodeling" (Fig. 2), could be provided by some analysis authors of NMJ innervation at an earlier time point(s) (P6, P7, P8)—of these data are available they should be presented.

In Fig. 5, what is the level of Tuba4a expression? From the graph it appears that Tuba4a is not present in spinal motor neurons. The role of Tuba4a in axon remodeling is also confusing – there are no changes in poly-innervated NMJs at P12-13, but changes are observed initially at P10-11. Showing NMJ innervation at earlier time points would be helpful (see above).

- The analyses of microtubule levels (e.g. Fig. 2a, but also elsewhere): authors should show the staining for α -tubulin to support this claim.

In Fig. 4, what is the level of polyglutamylation in dentate granule (DG) cells and in CA3 axons compared to IFT axons? Have authors confirmed that TTLL1 is expressed in DG neurons? A better representative image in panel c and/or showing additional examples from individual mice in the supplement would be helpful.

Does CCP1 KO rescues TTLL1 KO phenotype in NMJs remodeling? What is the role of CCP1/6 in pruning in the CNS?

In Fig. 8, what is the level of poly-innervated NMJs in this experiment? Further, for this activity experiment it is clear that polyglutamylation is influenced by neurotransmission, but the effect of activity on MT PTMs here is quite indirect (dependent on inference in the context of SpastKO). Have the authors considered assessment of neural activity effects in the context of heterozygosity of the mutants that they show here affect polyglutamylation?

Minor issues

The low polyE levels in CCP1/6 KO are suggested by the authors to indicate that “deglutamylases might be more relevant later in life” (line 211). It would help to clarify how this relates to their effect on on NMJsaxon remodeling?

What are the TTLL1 conditional mutants used in analysis in the Figure 4, and what controls were used in this experiment. Which Cre-line was used to recombine TTLL1 floxed allele?

Why were CCP1/6 KO mice used if only CCP1 is expressed in developing spinal motor neurons?—redundancy is cited, but additional clarification would be helpful.

Fig. 4g – The authors present quantification of “n>30 spines from 6 animals” and “n>43 spines from 8 animals”. It seems that they may have meant “n>30 dendrites” etc.. Please clarify, and also present how many dendrites were quantified per animal.

Scale bars in images showing corresponding experimental/control groups in Figure 4 and Figure 6j have different sizes, please correct or clarify...it would be best to have one scale bar/experiment and then indicate what it represents in the multiple panels for that experiment.

-text says “seeder glutamylaseses” but Fig. 1c shows “initiator”—best to choose one for consistency.

Reviewer #2

(Remarks to the Author)

The manuscript by Gavoci et al shows that tubulin polyglutamate chain elongation controls the postnatal remodelling of axons in motoneurons and in neurons of the central nervous system. They further show that, in response to neurotransmission, polyE elongation under the control of TTLL1 and the subsequent polyglutamylation of alpha-4 tubulin are key determinants of spastin-mediated pruning.

Overall, this study was well conducted and very clearly written. The results are convincing and bridge the gap between the princeps demonstration that polyglutamylation stimulates spastin activity on microtubules in vitro and in cells (Lacroix et al., 2010), and the physiological control this mechanism exerts in animal models to ensure the remodelling of axon targeting.

I have no major criticism about the experimental approaches or about the logic of the story. Nevertheless I want to raise a few remarks and questions.

1- I think that the above finding by Lacroix et al deserves a better visibility in the manuscript. I don't understand why it was barely cited by the authors, only in the discussion, along with later publications and why it did not appear in line 47 in the introduction section.

2- I understand that changing polyE chains on tubulin will affect microtubule dynamics and hence their growth, but I do not clearly see how the measurements made with EB3 comets are connected with this study and what they bring to its message. Maybe consider dropping them.

3- Given the time scale of axon pruning in neuromuscular junctions depicted in figure 3g, would it be possible to show images like those in figure 2i, in which a differential polyE signal could be detected between multiple axons, ideally with one axon exhibiting a low polyE level and the others a higher level? This would reinforce the authors findings and provide another strong argument that physiological differences such as neurotransmission effectiveness initiate the remodelling process.

4- Why did the authors show in all their quantifications half a violin plot with a nearby bar to depict the mean and s.e.m values instead of a full violin plot with just 3 bars inside? There should be a way to show all the experimental points within the violin plots. This will make the readers' interpretation easier and will compensate the truncation of most of the violin plots that result from splitting the y axes of the graphs.

5- In figure 1 e, the CCP6 curve is obviously missing (there is no curve with 28 percent increase as stated in the main text). What does the bottom (almost flat) curves represent?

6- In supplemental figure 2 b,f,j, the y axes are the distributions of comet lengths, not their densities.

Version 1:

Reviewer comments:

Reviewer #1

(Remarks to the Author)

I have read over the authors' rebuttal, considered their arguments for points they acknowledge but for which they think additional experiments are not warranted, and also looked over their experimental additions. Overall the authors have done a nice job of addressing my concerns with some new experiments, clarifications, and reasonable explanations for what they think would not be helpful to address with new work. Overall this study now strikes me as a nice contribution to the field and I have no further concerns.

Reviewer #2

(Remarks to the Author)

In its revised version, the authors correctly addressed the points I made with the first version of the article. They corrected some figures and took into account my remarks on data representation. I also noticed that data normalization has been better explained throughout the main text. In this context, the authors mentioned in line 109 an "anti-alpha-tubulin antibody recognizing all alpha-tubulin isotypes". When I went into the methods section, I could not find a reference for this antibody. The authors instead added a reference to Abcam's ab190573, which is an antibody against the alpha-4 tubulin isotype (line 478) and that was indeed missing from the first version. Please correct these points before publication. Regarding my point #2, I understand the arguments to keep the EB3 data and I am OK with this decision.

REVIEWER COMMENTS

Reviewer #1 (Remarks to the Author):

In this study, the authors address the microtubule (MT) posttranslational modification (PTM) polyglutamination in the context of motor axon and central nervous system axon pruning. They employ mutants in genes encoding seeder/initiator and elongator glutaminases (TTLL7 and TTLL1, respectively), and also deglutaminases (CCP1 and 6) that they show here via transcriptomic profiling are expressed in motor neurons over developmental timepoints that encompass motor axon pruning. The central results show that loss of TTLL function can delay neuromuscular synapse elimination and axon pruning, whereas loss of CCP function has the opposite effect. Additional work investigates the role played by the MT severing protein spastin in regulating MT polyglutamination and motor axon pruning, and also an initial assessment of neural activity's role in MT polyglutamination levels. Overall this is an interesting study that addresses a significant issue and demonstrates a novel role for MT PTMs in remodeling and axon pruning in at the NMJ, and in less detail in pruning of the hippocampal infrapyramidal tract (IFT). Attention to the points raised below, mostly minor, would strengthen this study.

We thank the Reviewer for his positive assessment of our study and address their points below.

R1.1: Weaknesses

data normalization: there are multiple examples in immunohistochemistry experiments where authors do not show fluorescence channel for signal normalization (e.g. neurofilaments). Authors also normalize polyglutamylation signals to β 3-tubulin, but this modification is mostly prominent on α -tubulins.

We appreciate the Reviewer's comments; this issue should have been clarified. The data that we present are normalized to a "structural" channel: For measurements of tubulin content, typically using neurofilament, which is not affected during axon pruning as we showed previously (Brill et al., 2016) or a "tubulin" channel (for measurements of posttranslational tubulin modifications) to compensate for differences in axon morphology, tissue depth and resulting light scatter, antibody penetration, etc. Indeed, for succinctness, we did not depict these reference immunostainings because we have used this approach for a while (Brill et al., 2016; Metzner et al., 2022). Still, we understand that this might lead to confusion, and we now depict the "normalization" channels in supplementary figures (**Extended data, Figure 2, 5, 8, 9, and 13**).

Further, the Reviewer is correct in pointing out that posttranslational tubulin modifications would ideally be normalized to the alpha-tubulin (Tuba) isoforms, as polyglutamyl side chains are predominantly attached to this part of the microtubule. We used normalization to the Tubb3 subunit as compensation for axons' microtubule content rather than to support a specific claim about the modified fraction of tubulin subunits. In our previous work (Brill et al., 2016), we verified the microtubule cytoskeleton's global breakdown in retreating motor axons on the ultrastructural level (Brill et al., Figure 3). On the light microscopic level, we proved a parallel decrease in Tubb3 and Tuba levels in pruning axons (Brill et al., 2016, Suppl. Figure 2). Thus, we are confident that the neuron-specific Tubb3 measurement is a suitable normalization factor for posttranslational tubulin modification levels. This normalization form is, in our hands, the most reliable from a technical point of view (neuronal specificity, expression level, direct fluorophore coupling, and penetration). Thus, based on a decade of experience with this form of quantification and our previous data detailed above, we are confident that Tubb3 immunostaining in our protocols reliably reflects overall microtubule mass. Still, to address this comment (see also **R.1.4**), we corroborate that Tubb3 and Tuba staining levels run parallel in TTLL1 and CCP1&6 mutants (**Figures 2 and 6**).

R1.2: Major comments

In Fig. 1d-f, TTLL1 is seen to be high throughout the development (P5 – P14), both TTLL7 and TTLL11 are observed to be increasing, and also CCP1 is also over postnatal timepoints. One might expect that if TTLLs are involved in pruning, their levels might increase without a concomitant increase in CCP. The authors show that in the TTLL1mnKO (Fig. 2j), there is already a high level of poly-innervated NMJs at P8-9. In line with this observation, the CCP1/6 cKO has fewer poly-innervated NMJs already at P7-8 (Fig. 6i). Together, this suggests that TTLL/CCPs participate significantly in the initial formation of the NMJ. Support for the notion that “Genetic ablation of TTLL1 delays motor axon remodeling” (Fig. 2), could be provided by some analysis authors of NMJ innervation at an earlier time point(s) (P6, P7, P8)—of these data are available they should be presented.

The Reviewer raises an important point: the CCPs and TTLLs certainly do more than regulate developmental pruning. The specific suggestion is that (de)glutamylases could be involved in NMJ formation, implying that motor neuron (MN)-specific TTLL1 KO and CCP1&6 KO muscles could show NMJ abnormalities at time points preceding the final elimination phase that we study. We agree with this concern but were confident we would have noticed any overt NMJ phenotype. Moreover, our intensive prior analysis of NMJ formation of spastin KO mice revealed no abnormalities (Brill et al. 2016, Suppl. Figures 5 and 6).

In line with these published data, we now provide a quantitative assessment of NMJ development, which proves to not be overtly affected in constitutive TTLL1^{KO}, Tuba4a^{KI}, and CCP1&6^{mnKO} (**Extended data, Figure 4**). We measured (1) the width of the endplate band (a measure of excessive sprouting), (2) the NMJ size (a measure of proper synapse maturation), and (3) the muscle fiber thickness (a measure of activity and muscle health). In addition, we also provide two new data sets for the number of polyinnervated NMJ at earlier time points (namely, P6-7 for TTLL1^{KO}, **Figure 2**; P6-7 and 8-9 for Tuba4a^{KI}, **Figure 5**).

R1.3: *In Fig. 5, what is the level of Tuba4a expression? From the graph it appears that Tuba4a is not present in spinal motor neurons. The role of Tuba4a in axon remodeling is also confusing – there are no changes in poly-innervated NMJs at P12-13, but changes are observed initially at P10-11. Showing NMJ innervation at earlier time points would be helpful (see above).*

Apologies here from our side. The graph had a typo: tuba4c = tuba4a, which we corrected. Thus, tuba4a RNA is present in motor neuron somata, although much less than other tubulin isoforms, e.g., tuba1a, tuba1b, and tuba1c.

Furthermore, we agree with the Reviewer’s assessment that adding an earlier time-point will make the tuba4a^{KI} data more conclusive and provide data on the NMJ innervation at P6-7 and 8-9 (**Figure 5**; see **R1.2** above). It might be noteworthy that synapse elimination is a very robust phenomenon. Hitherto, no manipulation that permanently retains NMJ polyinnervation is known, not even a genetic block of neurotransmission, despite the consensus that the driving force is “activity-dependent” competition (Buffelli et al., 2003). This also applies to our previous work: Neither genetically deleting what we now see as the “reader” during the synapse elimination process (i.e., spastin) nor pharmacological block of its down-stream effects (i.e., microtubule breakdown) permanently maintained polyinnervation (Brill et al., 2016). Moreover, while polyglutamylation is enriched on Tuba4a (Hausrat et al., 2022), it is not exclusive to this tubulin isoform (Maliekal et al., 2022). Thus, compensatory action via other alpha-tubulin (Tuba) isoforms seems possible in Tuba4a^{KI} mice but is experimentally challenging to address.

R1.4: *The analyses of microtubule levels (e.g. Fig. 2a, but also elsewhere): authors should show the staining for α -tubulin to support this claim.*

See also **R.1.1**. To this, we performed pan-alpha-tubulin immunostainings in several of our mutant mice to corroborate that Tubb3 and Tuba staining levels change similarly (**Figures 2 and 6**).

R1.5: In Fig. 4, what is the level of polyglutamylation in dentate granule (DG) cells and in CA3 axons compared to IFT axons? Have authors confirmed that *TTL1* is expressed in DG neurons?

The Reviewer correctly pointed out that we have not demonstrated the expression of *TTL1* or the polyglutamylation level in DG granule cells.

Direct immunostaining of *TTL1* is not feasible, as we are unaware of a reliable *TTL1* antibody working in immunohistochemistry on brain slices. Still, a global in-situ hybridization assessment of *TTL1* expression in P56 mouse brains has been performed for the Allen Brain Atlas. We inserted a supplementary figure showing such in-situ hybridization data (**Extended data, Figure 6**). *TTL1* and *CCP1* showed widespread RNA expression in the CNS and robust expression in hippocampal DG cells, consistent with the observed pruning defects. In addition, our prior work has revealed hyperglutamylation by immunoblotting of hippocampi, cortices, and cerebella tissue extracts in *CCP1*^{6^{KO} and reduced polyE signal in *TTL1*^{KO} by immunoblotting cerebella extracts (Magiera et al., 2018); similarly, tubulin polyglutamylation levels affect organelle trafficking in hippocampal neurons in vitro (Bodakuntla et al., 2021), suggesting active polyglutamylation.}

Assessing polyE levels directly in the hippocampus proved similarly challenging. First, the PolyE antibody did not reliably work on our hippocampal paraffin sections. Thus, we could only analyze GT335 stainings (an antibody that recognizes glutamylation independent of glutamate chain length). Further, the stereotyped pruning of the IPB in mice takes around 20 days (between postnatal days 10-30; Bagri et al., 2003); compared to this extended period, the pruning process of single axon branches is likely short. Therefore, at any time, polyglutamylation is likely increased only in those IPB axons about to undergo pruning and not in all the IPB axons. To reliably analyze the levels of tubulin polyglutamylation in mossy fiber IPB and SPB, we would need to compare PolyE immunostainings in both bundles with single-axon resolution, which, due to resolution restrictions, we were not able to do given the dense neuropil (with intermingled axons and dendrites) in the hippocampal CA3 region. So, all we could do was stain calbindin D-28K to highlight the areas of IPB and SPB (trying to exclude intermingling dendrites; **Reviewer Figure 1a** and **1b**). We found no significant difference towards increased bulk tubulin glutamylation in the IPB (**Reviewer Figure 1c**). As this data could still be compatible with an increased polyE signal in some axon branches that are in the process of pruning, they remain inconclusive, and we chose to present this quantification only to the Reviewer.

Reviewer Figure 1

(a) Schematic of the hippocampus showing the suprapyramidal main bundle (SPB, yellow), the infrapyramidal bundle (IPB, magenta), as well as the dentate gyrus (DG), CA1, CA2, and CA3 regions.

(b) Confocal stacks of coronal brain sections of 2-week-old wildtype mice stained for Calbindin D-28k (right) and GT335 (left). The magenta ROI depicts the area of GT335 intensity measurement based on the Calbindin staining; the yellow ROI depicts the area of GT335 intensity measurement in SPB.

(c) Quantification of GT335 immunostaining intensity in SPB and IPB (normalized on average GT335 intensity of both bundles).

Scale bar, 100 μ m.

R1.6: *A better representative image in panel c and/or showing additional examples from individual mice in the supplement would be helpful.*

We agree with the Reviewer. We selected a more representative image (**Figures 4 and 7**) and included all the quantified IPBs in the supplement (**Extended data, Figures 7 and 11**).

R1.7: *Does CCP1 KO rescue TTLL1 KO phenotype in NMJs remodeling?*

While we agree that this could be an interesting experiment, we would argue that it is not key to our paper's message. Animal availability and regulations make this a complex experiment that we did not pursue experimentally for the following reasons: The crossbreeding of CCP1^{KO} and TTLL1^{KO} is not available to us currently, and the breeding program, as well as prior required acceptance of an amendment to our animal protocol would need many months. Moreover, we have shown in the cerebellum that the genetic deletion of TTLL1 rescued CCP1 loss (Magiera et al., 2018). We have now edited the discussion, pointing out that in analogy with the cerebellum, a balanced action of TTLLs and CCPs could be key to spastin's activity.

R1.8: *What is the role of CCP1/6 in pruning in the CNS?*

We addressed this question with a new experiment. As constitutive CCP1&CCP6 double KOs are not born at Mendelian ratios, breeding would take months and needs approval (see above). So, we are not in a position to test this question experimentally. However, to approach it, in revision, we analyzed hippocampal pruning in CCP1^{KO} at postnatal day 14. We found accelerated pruning (**Figure 7; Extended data, Figure 11**), indicating that deglutamylases also play a role in CNS pruning.

R1.9: *In Fig. 8, what is the level of poly-innervated NMJs in this experiment? Further, for this activity experiment it is clear that polyglutamylation is influenced by neurotransmission, but the effect of activity on MT PTMs here is quite indirect (dependent on inference in the context of SpastKO). Have the authors considered the assessment of neural activity effects in the context of heterozygosity of the mutants that they show here affect polyglutamylation?*

In the first part of this comment, the Reviewer asks for the number of polyinnervated NMJs following blockage of neurotransmission. The effect of blocking NMJ activity on synapse elimination is well-studied (Buffelli et al., 2003). Indeed, we have published previously that alpha-bungarotoxin injection significantly delays the axonal remodeling process (controls ~20% doubly innervated NMJs vs ~30% upon blockage of neurotransmission ((Wang et al., 2021), Figure 4D). In the revision, we illustrate the loss of polyE signal in pruning axons by adding a new micrograph (**Figure 9a**).

The Reviewer also asked what would happen if we combined partial blocking activity with incomplete (heterozygous) spastin deletion. However, any outcome of this experiment would be compatible with our model. So, while this is certainly an interesting experiment, we feel it might not be key – and given its invasiveness for the experimental animals, we chose not to do it.

Minor issues

R1.10: *The low polyE levels in CCP1/6 KO are suggested by the authors to indicate that “deglutamylases might be more relevant later in life” (line 211). It would help to clarify how this relates to their effect on NMJs axon remodeling?*

The Reviewer questions a statement in the context of our (unexpected) observation that upon CCP1&6 deletion, polyglutamylation levels only rise in adult animals. At the same time, co-deletion of CCP1 and spastin leads to increased polyglutamylation. We understand that our argument might be seen as oversimplified and unnecessary, as the steady-state levels of polyglutamylation are as much the result of the activity of enzymes that mediate polyglutamylation (CCP vs. TTLL)

as others that remove the product of such enzymatic reactions (spastin). Thus, in our revision, we removed this statement.

R1.11: *What are the TTLL1 conditional mutants used in analysis in the Figure 4, and what controls were used in this experiment. Which Cre-line was used to recombine TTLL1 floxed allele?*

These constitutive TTLL1KO mice were obtained by crossbreeding PGK-Cre to TTLL1 conditional (flox) animals. Homozygous TTLL1^{KO} were experimental animals, and controls were homozygous TTLL1 wildtype littermate controls (TTLL1 wildtype, WT), resulting from a TTLL1 heterozygous (TTLL1^{KO/WT} X TTLL1^{KO/WT}) crossbreeding. We added this information to the Experimental Procedures.

R1.12: *Why were CCP1/6 KO mice used if only CCP1 is expressed in developing spinal motor neurons?—redundancy is cited, but additional clarification would be helpful.*

The Reviewer is, of course, correct; the logic of our argument would dictate to first examine CCP1^{KO} and only resort to double KOs if there were hints of compensation. Using the CCP1&6 double KOs was a matter of risk mitigation: CCP6—albeit at much lower levels than CCP1—is expressed in motor neurons. Given that we had seen some level of compensation by CCP6 after the deletion of CCP1 in prior work in the CNS (Magiera et al., 2018), we decided to delete both alleles. This approach allowed us to maximize the chances of a measurable phenotype and minimized the number of “unused” animals dictated by our local animal welfare authorities.

R1.13: *Fig. 4g – The authors present quantification of “n>30 spines from 6 animals” and “n>43 spines from 8 animals”. It seems that they may have meant “n>30 dendrites” etc. Please clarify, and also present how many dendrites were quantified per animal.*

This was misleading. Indeed, we quantified dendritic branches. In the revision, we corrected it accordingly and stated the number of branches in the figure legend (**Figures 4 and 7**).

R1.14: *Scale bars in images showing corresponding experimental/control groups in Figure 4 and Figure 6j have different sizes, please correct or clarify...it would be best to have one scale bar/experiment and then indicate what it represents in the multiple panels for that experiment.*

We followed the Reviewer’s suggestion and presented only one scale bar in the revised version.

R1.15: *Text says “seeder glutamylases” but Fig. 1c shows “initiator”—best to choose one for consistency.*

We edited the manuscript and consistently used “initiator glutamylase”.

Reviewer #2 (Remarks to the Author):

The manuscript by Gavoci et al shows that tubulin polyglutamate chain elongation controls the postnatal remodelling of axons in motoneurons and in neurons of the central nervous system. They further show that, in response to neurotransmission, polyE elongation under the control of TTL1 and the subsequent polyglutamylation of alpha-4 tubulin are key determinants of spastin-mediated pruning. Overall, this study was well conducted and very clearly written. The results are convincing and bridge the gap between the princeps demonstration that polyglutamylation stimulates spastin activity on microtubules in vitro and in cells (c), and the physiological control this mechanism exerts in animal models to ensure the remodelling of axon targeting. I have no major criticism about the experimental approaches or about the logic of the story. Nevertheless I want to raise a few remarks and questions.

We also thank this Reviewer for their positive feedback on our manuscript. Responding to their suggestions, we implemented the changes detailed below.

R2.1: *I think that the above finding by Lacroix et al deserves a better visibility in the manuscript. I don't understand why it was barely cited by the authors, only in the discussion, along with later publications and why it did not appear in line 47 in the introduction section.*

The Reviewer is right; we emphasize this important prior work and refer to it in line 46.

R2.2: *I understand that changing polyE chains on tubulin will affect microtubule dynamics and hence their growth, but I do not clearly see how the measurements made with EB3 comets are connected with this study and what they bring to its message. Maybe consider dropping them.*

We see the Reviewer's point and could move these data to another small paper; still, when publishing prior work, we have learned that many microtubule researchers find this analysis of microtubule dynamics informative; we also believe that in certain situations, when combined with qualifications of microtubule mass, the EB3 measurements can enlighten the mechanism of severing (see Brill et al., 2016 for a more detailed discussion of this point). Thus, we decided to keep the EB3 data after editorial consultation with Dr. Morales.

R2.3: *Given the time scale of axon pruning in neuromuscular junctions depicted in figure 3g, would it be possible to show images like those in figure 2i, in which a differential polyE signal could be detected between multiple axons, ideally with one axon exhibiting a low polyE level and the others a higher level? This would reinforce the authors findings and provide another strong argument that physiological differences such as neurotransmission effectiveness initiate the remodelling process.*

We agree that such an illustration might make a key starting point of our study immediately apparent to the readers. We had omitted it because, in prior published work, we showed a clear quantitative relationship of the polyglutamylation level in terminal motor axons depending on the synaptic occupancy ((Brill et al., 2016), Figure 5 G). However, the Reviewer is correct that a repeated visualization of this key result is essential here. We now inserted a micrograph of a doubly innervated NMJ with a substantial disparity of the axons (one > 80% of the synaptic territory, a likely "winner," the counterpart with < 20%) and distinct polyglutamylation levels (**Figure 9a**) and mention this in the results section.

R2.4: *Why did the authors show in all their quantifications half a violin plot with a nearby bar to depict the mean and s.e.m values instead of a full violin plot with just 3 bars inside? There should be a way to show all the experimental points within the violin plots. This will make the readers' interpretation easier and will compensate the truncation of most of the violin plots that result from splitting the y axes of the graphs.*

We believed this to be an efficient way of representing the distribution and the effect size, but we understand the objection. Moreover, in our spread data, the pure scatter representation drowns out the effects, and if there are more than 25 data points, the actual local data density is hard to judge. At the same time, outliers get overemphasized as the central data cloud saturates. We now show half violins (cut at the 90% level of the dominant violin to avoid overrepresentation of outliers), box plots, and all single data points in an opaque line representing the single data points – this, we believe, gives the reader full autonomy in deciding, which aspect of the data to focus on.

R2.5: *In figure 1 e, the CCP6 curve is obviously missing (there is no curve with 28 percent increase as stated in the main text). What does the bottom (almost flat) curves represent?*

The Reviewer pointed out correctly (see also Reviewer 1, **R1.12**) that we forgot to label all curves on the CCP graph. To clarify this, we added an expanded and adequately labeled view of the CCP2-CCP6 RiboTag curves (**Figure 1**).

R2.6: *In supplemental figure 2 b,f,j, the y axes are the distributions of comet lengths, not their densities.*

Thanks to the Reviewer for picking this up! We corrected the y-axis of the now renumbered **Extended data, Figure 3b, f, j**.

References

- Bagri, A., H.J. Cheng, A. Yaron, S.J. Pleasure, and M. Tessier-Lavigne. 2003. Stereotyped pruning of long hippocampal axon branches triggered by retraction inducers of the semaphorin family. *Cell*. 113:285-299.
- Bodakuntla, S., X. Yuan, M. Genova, S. Gadadhar, S. Leboucher, M.C. Birling, D. Klein, R. Martini, C. Janke, and M.M. Magiera. 2021. Distinct roles of alpha- and beta-tubulin polyglutamylation in controlling axonal transport and in neurodegeneration. *EMBO J*. 40:e108498.
- Brill, M.S., T. Kleele, L. Ruschkies, M. Wang, N.A. Marahori, M.S. Reuter, T.J. Hausrat, E. Weigand, M. Fisher, A. Ahles, S. Engelhardt, D.L. Bishop, M. Kneussel, and T. Misgeld. 2016. Branch-Specific Microtubule Destabilization Mediates Axon Branch Loss during Neuromuscular Synapse Elimination. *Neuron*. 92:845-856.
- Buffelli, M., R.W. Burgess, G. Feng, C.G. Lobe, J.W. Lichtman, and J.R. Sanes. 2003. Genetic evidence that relative synaptic efficacy biases the outcome of synaptic competition. *Nature*. 424:430-434.
- Hausrat, T.J., P.C. Janiesch, P. Breiden, D. Lutz, S. Hoffmeister-Ullerich, I. Hermans-Borgmeyer, A.V. Failla, and M. Kneussel. 2022. Disruption of tubulin-alpha4a polyglutamylation prevents aggregation of hyper-phosphorylated tau and microglia activation in mice. *Nature communications*. 13:4192.
- Magiera, M.M., S. Bodakuntla, J. Ziak, S. Lacomme, P. Marques Sousa, S. Leboucher, T.J. Hausrat, C. Bosc, A. Andrieux, M. Kneussel, M. Landry, A. Calas, M. Balastik, and C. Janke. 2018. Excessive tubulin polyglutamylation causes neurodegeneration and perturbs neuronal transport. *EMBO J*. 37.
- Maliekal, T.T., D. Dharmapal, and S. Sengupta. 2022. Tubulin Isotypes: Emerging Roles in Defining Cancer Stem Cell Niche. *Front Immunol*. 13:876278.
- Metzner, K., O. Darawsha, M. Wang, N. Gaur, Y. Cheng, A. Rodiger, C. Frahm, O.W. Witte, F. Perocchi, H. Axer, J. Grosskreutz, and M.S. Brill. 2022. Age-dependent increase of cytoskeletal components in sensory axons in human skin. *Front Cell Dev Biol*. 10:965382.
- Wang, M., T. Kleele, Y. Xiao, G. Plucinska, P. Avramopoulos, S. Engelhardt, M.H. Schwab, M. Kneussel, T. Czopka, D.L. Sherman, P.J. Brophy, T. Misgeld, and M.S. Brill. 2021. Completion of neuronal remodeling prompts myelination along developing motor axon branches. *J Cell Biol*. 220.

REVIEWER COMMENTS

Reviewer #1 (Remarks to the Author):

I have read over the authors' rebuttal, considered their arguments for points they acknowledge but for which they think additional experiments are not warranted, and also looked over their experimental additions. Overall the authors have done a nice job of addressing my concerns with some new experiments, clarifications, and reasonable explanations for what they think would not be helpful to address with new work. Overall this study now strikes me as a nice contribution to the field and I have no further concerns.

We thank the Reviewer for their very positive assessment of our revised study.

Reviewer #2 (Remarks to the Author):

In its revised version, the authors correctly addressed the points I made with the first version of the article. They corrected some figures and took into account my remarks on data representation. I also noticed that data normalization has been better explained throughout the main text.

In this context, the authors mentioned in line 109 an "anti-alpha-tubulin antibody recognizing all alpha-tubulin isotypes". When I went into the methods section, I could not find a reference for this antibody. The authors instead added a reference to Abcam's ab190573, which is an antibody against the alpha-4 tubulin isotype (line 478) and that was indeed missing from the first version. Please correct these points before publication.

Regarding my point #2, I understand the arguments to keep the EB3 data and I am OK with this decision.

We also thank this Reviewer for their positive assessment of our revised study.

In the manuscript, we have now corrected the catalogue number of the alpha-tubulin antibody in the Methods section. Accordingly, we removed the reference to the alpha-tubulin 4a antibody, as we did not conduct immunostaining experiments for the tuba4a subunit.